# Repairing Systematic Outliers by Learning Clean Subspaces in VAEs

## Abstract

Data cleaning often comprises outlier detection and data repair. Systematic errors result from nearly deterministic transformations that occur repeatedly in the data, e.g. specific image pixels being set to default values or watermarks. Consequently, models with enough capacity easily overfit to these errors, making detection and repair difficult. Seeing as a systematic outlier is a combination of patterns of a clean instance and systematic error patterns, our main insight is that inliers can be modelled by a smaller representation (subspace) in a model than outliers. By exploiting this, we propose *Clean Subspace Variational Autoencoder (CLSVAE)*, a novel semi-supervised model for detection and automated repair of systematic errors. The main idea is to partition the latent space and model inlier and outlier patterns separately. CLSVAE is effective with much less labelled data compared to previous related models, often with less than 2% of the data. We provide experiments using three image datasets in scenarios with different levels of corruption and labelled set sizes, comparing to relevant baselines. CLSVAE provides superior repairs without human intervention, e.g. with just 0.25% of labelled data we see a relative error decrease of 58% compared to the closest baseline.

## 1 Introduction

Often practitioners have to deal with dirty datasets before they can start applying machine learning (ML) models. We focus on datasets where some instances have been corrupted by noise, producing outliers. The corresponding clean instances prior to corruption are called inliers, as well as any other instances that have not been corrupted. Because the presence of outliers can degrade the performance of ML methods (Krishnan et al., 2016; Liu et al., 2020), a standard option is to resort to a data cleaning pipeline before applying any model. This pipeline includes two key tasks: i) *outlier detection* (Ruff et al., 2021), detecting all outliers; ii) *repair* those outlier instances (Neutatz et al., 2021; Wan et al., 2020), recovering the underlying inlier instance. Ultimately, the goal is to propose a method that performs these steps automatically, i.e. *automatic detection and repair*.

Generally, we can consider two types of errors present in an outlier: *random* or *systematic* (Taylor, 1997; Liu et al., 2020). Random errors corrupt each instance independently, and feature value changes are sampled from an unknown distribution. This noising cannot be replicated in a repeatable manner. For continuous features, a common example of this type of error are those well-modelled by additive noise with zero-mean. Systematic errors result from deterministic, or nearly deterministic, transformations that occur repeatedly in the data. Examples of systematic errors include watermarks or deterministic pixel corruption (e.g. artifacts) in images; additive offsets or replacement by default values (e.g. NaN) in sensor data; deterministic change of categories (mislabelling) or of name formats in categorical features in tabular data. Usually, the same features are affected, but not always. In most cases, this noising can be replicated.

These two types of errors have a different impact when it comes to models performing detection or repair. Random errors do not show a distinct pattern across outliers and thus are not predictable. As a result, unsupervised models using regularization or data reweighting (Zhou & Paffenroth, 2017; Akrami et al., 2019; Eduardo et al., 2020) can avoid overfitting to random errors. On the contrary, a systematic error shows a pattern across outliers, as a result of the (nearly) deterministic transformation, making them predictable (Liu et al., 2020). This property makes higher capacity models (e.g. deep learning) prone to overfitting to these errors, even in the presence of regularization. As a re-

sult, models for outlier detection and repair in the presence of systematic errors more easily conflate outliers with inliers. In this work we study how to develop a model for data cleaning robust to the effect of systematic errors.

One solution is to provide some supervision, so the model can distinguish between inliers and outliers with systematic errors. This supervision can be provided in different forms, such as logic rules (Rekatsinas et al., 2017) or programs (Lew et al., 2021) describing the underlying clean data. However, these may require expert knowledge or substantial effort to formalize. Hence, it is easier and less time consuming for the practitioner to simply provide a *trusted set* as form of supervision. A trusted set is a small labelled subset of the data, which can be used to train a method for detection and repair. The user labels the instances either inlier or outlier, providing a few examples (e.g. 10 instances) per type of systematic error to repair. Overall, this might correspond to less than 2% of the entire dataset (*sparse semi-supervision*). Equally important, labelling does not require the user to manually repair the instances in the trusted set.

Deep generative models (DGM) have high capacity (flexible) and thus can easily overfit to systematic errors. This motivates us to propose a method for detection and repair of such errors based on a semi-supervised DGM, which to the best of our knowledge has not been explored. We postulate that inlier data needs a *smaller representation* relative to outlier data when being represented by DGMs which we call the *compression hypothesis*. In addition, we claim that outliers are a *combination* of a *representation describing the inlier portion* of an instance, and a *representation describing the type of systematic error*.

Given these insights, we propose Clean Subspace Variational Autoencoder (CLSVAE), a novel semi-supervised model for detection and automated repair of systematic errors. This model deviates from a standard VAE (Kingma & Welling, 2014), in two ways. First, the latent representation is partitioned into two subspaces: one that describes the data if it were an inlier (clean subspace), and the other that describes what systematic error (if any) has been applied (dirty subspace). The model is encouraged to learn a disentangled representation using a simple yet effective approach: for outliers the decoder will use a clean subspace concatenated with the dirty subspace; in turn, for inliers the decoder will reuse the same clean subspace but concatenated with random noise. Then, at repair time the decoder will only need the clean subspace to reconstruct the underlying inlier. Secondly, we introduce semi-supervision through a trusted set, in so helping the model distinguish between inliers and outliers with systematic errors. Additionally, to encourage the clean subspace to represent inlier data, and the dirty subspace to represent systematic errors, we introduce a novel penalty term minimizing their mutual information (MI). This penalty is based on the *distance correlation* (DC) (Székely et al., 2007), and it improves model performance (stability) and repair quality. Compared to baselines we provide superior repairs, particularly, we show significant advantage in smaller trusted sets or when more of the dataset is corrupted.

## 2 RELATED WORK

For random errors, several works have explored detection (Akrami et al., 2019; Ruff et al., 2019; Liu et al., 2020; Lai et al., 2019). Some have proposed methods for detection and automated repair of random errors (Eduardo et al., 2020; Zhou & Paffenroth, 2017; Krishnan et al., 2016). A few works have explored outlier detection for systematic outliers, both semi-supervised (Ruff et al., 2019) and unsupervised (Liu et al., 2020; Lai et al., 2020), but these works do not consider automated repair.

For tabular data, methods have been proposed that can do repair after detection has been performed; for an overview, see (Ilyas & Chu, 2019; Chu et al., 2016). These use probabilistic models melded with logic rules, i.e. probabilistic relational models (Rekatsinas et al., 2017), or user-written programmatic descriptions of the data, e.g. in a probabilistic programming language (Lew et al., 2021). They have the potential to capture systematic errors. However, these require the user to provide rules or programs that characterize clean data, which requires manual effort and expertise. In contrast, labelling a few inlier and outlier instances to build a trusted set may be more user-friendly.

The idea of using unsupervised models for outlier detection is not new (Schölkopf et al., 1999; Liu et al., 2008). Unsupervised removal from a dataset of a small fraction of instances suspected of being outliers has been explored in Koh et al. (2018); Diakonikolas et al. (2018); Liu et al. (2020), when used against adversarial errors it is called *data sanitization* (Koh et al., 2018). However, if

one wants to repair existing dirty data, a common unsupervised approach for moderate corruption is to apply enough regularization or data reweighting to an autoencoder (Eduardo et al., 2020; Zhou & Paffenroth, 2017; Akrami et al., 2019), hopefully repairing the errors at reconstruction. This has proven successful for random errors, though as we show, this is less effective for systematic errors. Often regularization is too strong leading to bad repair quality, since reconstruction is collapsing to mean behaviour – e.g. blurry image samples, missing details.

*Attribute manipulation* models (Klys et al., 2018; Choi et al., 2020) usually rely on extensive labelled data, usually fully supervised. Data cleaning can be seen as an attribute manipulation problem with one attribute: either the instance is an inlier or outlier. Some models like CVAE (Sohn et al., 2015) may use discrete latent variables, instead of continuous, to model attributes. These may lack the capacity to capture diversity in the same attribute (Joy et al., 2020), e.g. distinct types of systematic errors in the data, and thus offer a poor fit to the data. Later, this may result in poor repair.

*Disentanglement models*, unsupervised (Locatello et al., 2019; Tonolini et al., 2019) and semi-supervised (Ilse et al., 2020; Locatello et al., 2019; Joy et al., 2020), encourage individual (continuous) latent variables to capture different attributes of the data instances. In theory, a disentanglement model could isolate the inlier / outlier attribute into a latent variable, then after training one could find a value for this variable that repairs the outlier. In practice, this is much simpler with semi-supervised models, since we know which variable corresponds the attribute, but these tend to use more labelled data than we consider. In either case, these models usually need additional processing, often requiring human in the loop to explore the latent variable and find the best repair. Conversely, once our model is trained, further human intervention is not needed, so that repair is automated.

## 3 PROBLEM DEFINITION

We assume the user knows that systematic errors have corrupted dataset $\mathcal{X}$, and thus outliers exist therein, but the inliers are still the majority. The user also has an idea of what patterns constitute systematic errors, and is able to recognize them. Intuitively, we think of inlier data being characterized by a set of patterns, which we call *clean patterns*, and those that constitute (systematic) errors, called *dirty patterns. An outlier is an instance that has been corrupted by systematic errors, where prior to that would be constituted by clean patterns only*. Formally, for $\tilde{x}$ an underlying inlier instance, an outlier is defined by $x = f_{cr}(\tilde{x})$. We define $f_{cr}$ as a *general transformation that corrupts the inlier instance with systematic errors*, and not necessarily invertible. Each type of systematic error usually affects specific features of $\tilde{x}$ in the same predictable way, repeatedly in the dataset. This is unlike random errors, which both the feature and changed values are at random throughout instances.

The user builds a small *trusted set* by labelling a few of the inliers and outliers in the train set $\mathcal{X}$, forming the labelled subset $\mathcal{X}_l = \{x_n\}_{n=1}^{N_l}$. So we have the dirty dataset $\mathcal{X} = \mathcal{X}_l \cup \mathcal{X}_u$, where $\mathcal{X}_u = \{x_n\}_{n=1}^{N_u}$ is the unlabelled part. The overall size of the train set is $N = N_l + N_u$. Each $x_n \in \mathcal{X}_l$ is associated with a label $y_n \in \{0, 1\}$, which indicates whether $x_n$ is an inlier ($y_n = 1$) or an outlier ($y_n = 0$). We write $\mathcal{Y}_l = \{y_n\}_{n=1}^{N_l}$ and thus the trusted set is formally defined by $(\mathcal{X}_l, \mathcal{Y}_l)$. The trusted set should be representative of the inliers and outliers in the data. Note that there is a set of different corrupting transformations, i.e. systematic error types, each applied to several instances. Hence, the trusted set should provide at least a few labelled samples per type of systematic error. This is important so as to help the model distinguish between inliers and outliers. In our problem, the labelled portion of the dataset $\mathcal{X}_l$ (trusted set) is significantly smaller than the unlabelled portion $\mathcal{X}_u$, e.g. 0.5% of $N_u$. Given how small the trusted set is, we refer to this as *sparse semi-supervision*.

In this work, the main task is to first perform *outlier detection* on $\mathcal{X}$, in order to discover $y$ for each instance $x$. The second task is *repair* those instances that are considered to be outliers ($y = 0$). Specifically, the model needs to provide a repair transformation $g_r$ such that $\hat{x} = g_r(x)$ where $\hat{x} \approx \tilde{x}$ (i.e. repair is close enough to the underlying inlier instance). Thus, we want a model that performs these two steps automatically, i.e. without a human in the loop.

## 4 OUR PROPOSAL: CLEAN SUBSPACE VAE (CLSVAE)

In this section, we introduce the generative and variational models for our proposal. Our Variational Autoencoder (VAE) model is motivated by three observations:

- ***Outliers are systematic.*** Outlier instances can be described by a number of predictable recurring patterns, some of which are considered dirty (e.g. black patch on an image). Our assumption is that the latter patterns can be well represented by a DGM.

- ***Compression hypothesis.*** Inlier data can be compressed further than outlier data. We assume that less capacity (parameters or variables) is needed by DGMs to represent inliers.

- ***Outliers are a combination of clean and dirty patterns.*** Because clean and dirty patterns produce different visible effects in the instance, the effects of clean and dirty patterns can be modelled separately. Hence, this DGM only requires the parameters (or variables) of the clean patterns to generate a repair.

We call our model the *sparse semi-supervised Clean Subspace VAE (CLSVAE)*. This reflects the fact that our model has a partitioned latent space: a subspace for clean patterns, and a subspace for dirty patterns. Thus at test time only the subspace for clean patterns is used to generate a repair. A standard VAE (Kingma & Welling, 2014), with weight decay, is provided in Annex A for readers unfamiliar with this type of model.

## 4.1 GENERATIVE MODEL

The main idea is that if the *compression hypothesis* holds inlier samples ($y = 1$) will have a smaller representation relative to outliers ($y = 0$), whilst outliers need an extra representation to model errors. In our model, $\boldsymbol{z}_c$ will represent inliers, $\boldsymbol{z}_d$ will represent the error pattern for outliers, and $\boldsymbol{z}_\epsilon$ is random noise. The overall latent code is $\boldsymbol{z} = [\boldsymbol{z}_c; y\, \boldsymbol{z}_\epsilon + (1 - y)\, \boldsymbol{z}_d]$, where $[\ ;\ ]$ defines vector concatenation, leading to a latent space that is partitioned into two subspaces. If generating an inlier ($y = 1$) then $\boldsymbol{z}_c \in \mathbb{R}^q$ is by itself responsible for modelling all the clean patterns present in an instance, encouraging a lower dimensional manifold for inlier data. This is due to $\boldsymbol{z}_\epsilon$ being a random noise vector of the same dimension as $\boldsymbol{z}_d \in \mathbb{R}^p$, devoid of additional information. If generating an outlier ($y = 0$) both $\boldsymbol{z}_c$ and $\boldsymbol{z}_d$ are used, where $\boldsymbol{z}_d$ models the different types of dirty patterns that can be detected in an instance. In fact, using $\boldsymbol{z}_\epsilon$ during inlier generation encourages $\boldsymbol{z}_d$ to model dirty patterns only, encouraging a higher dimensional manifold for outlier data.

Our generative model is therefore defined by the joint distribution between $y$ and latent subspaces $\boldsymbol{z}_c$ and $\boldsymbol{z}_d$. The main idea expressed above can be written as a two component mixture model for the decoder, see eq. (4), where $y$ is the gating variable. Hence we have

$$p_\theta(\boldsymbol{x}, \boldsymbol{z}_c, \boldsymbol{z}_d, \boldsymbol{z}_\epsilon, y) = p_\theta(\boldsymbol{x}|\boldsymbol{z}_c, \boldsymbol{z}_d, \boldsymbol{z}_\epsilon, y)p_{\sigma_c}(\boldsymbol{z}_c)p_{\sigma_d}(\boldsymbol{z}_d)p_{\sigma_\epsilon}(\boldsymbol{z}_\epsilon)p_\alpha(y), \tag{1}$$

where

$$p_\alpha(y) = \text{Bernoulli}(y|\alpha), \tag{2}$$

$$p_{\sigma_c}(\boldsymbol{z}_c) = \mathcal{N}(\boldsymbol{z}_c|\boldsymbol{0}, \sigma_c^2\boldsymbol{I}) \quad p_{\sigma_d}(\boldsymbol{z}_d) = \mathcal{N}(\boldsymbol{z}_d|\boldsymbol{0}, \sigma_d^2\boldsymbol{I}) \quad p_{\sigma_\epsilon}(\boldsymbol{z}_\epsilon) = \mathcal{N}(\boldsymbol{z}_\epsilon|\boldsymbol{0}, \sigma_\epsilon^2\boldsymbol{I}), \tag{3}$$

$$p_\theta(\boldsymbol{x}|\boldsymbol{z}_c, \boldsymbol{z}_d, \boldsymbol{z}_\epsilon, y) = p_\theta(\boldsymbol{x}\,|\,[\boldsymbol{z}_c;\ \boldsymbol{z}_\epsilon])^y p_\theta(\boldsymbol{x}\,|\,[\boldsymbol{z}_c;\ \boldsymbol{z}_d])^{(1-y)}, \tag{4}$$

where the density $p_\theta(\boldsymbol{x}\,|\,[\cdot;\ \cdot])$ is parameterized by a neural network. We assume inlier data has a smaller variance as whole than outlier data, so we use a $\sigma_c$ that is *much smaller than $\sigma_d$*. Note that $\boldsymbol{z}_\epsilon$ is just Gaussian random noise[1] with $\sigma_\epsilon < \sigma_d$. Hence, after training, the region around $\boldsymbol{0}$ (zero mean) for this subspace encourages $p_\theta(\boldsymbol{x}\,|\,[\cdot;\ \cdot])$ to only use $\boldsymbol{z}_c$ for reconstruction, obtaining a repaired instance. The parameter $\alpha$ reflects the prior belief on the fraction of clean data. Smaller values for $\alpha$ means more data points are rejected when modelling $\boldsymbol{z}_c$, which offers more robustness. So we have $\sigma_\epsilon$, $\sigma_c$, $\sigma_d$ and $\alpha$ as hyper-parameters.

## 4.2 VARIATIONAL MODEL

We consider separate encoders for $\boldsymbol{z}_c$ and $\boldsymbol{z}_d$, and make $y$ depend on $\boldsymbol{z}_c$ and $\boldsymbol{z}_d$. The idea is that parameters of each encoder focus on different aspects, i.e. clean or dirty patterns respectively. The model factorizes as

$$q(\boldsymbol{z}_c, \boldsymbol{z}_d, \boldsymbol{z}_\epsilon, y|\boldsymbol{x}) = q_{\phi_y}(y|\boldsymbol{z}_c, \boldsymbol{z}_d)q_{\phi_c}(\boldsymbol{z}_c|\boldsymbol{x})q_{\phi_d}(\boldsymbol{z}_d|\boldsymbol{x})q_{\sigma_\epsilon}(\boldsymbol{z}_\epsilon), \tag{5}$$

and

$$q_{\phi_c}(\boldsymbol{z}_c|\boldsymbol{x}) = \mathcal{N}(\boldsymbol{z}_c|\boldsymbol{\mu}_{\phi_c}(\boldsymbol{x}), \boldsymbol{\sigma}_{\phi_c}^2(\boldsymbol{x})), \quad q_{\phi_d}(\boldsymbol{z}_d|\boldsymbol{x}) = \mathcal{N}(\boldsymbol{z}_d|\boldsymbol{\mu}_{\phi_d}(\boldsymbol{x}), \boldsymbol{\sigma}_{\phi_d}^2(\boldsymbol{x})), \tag{6}$$

---

[1] We tried using $\boldsymbol{z}_\epsilon = \boldsymbol{0}$ at training, however this did not work as well as setting $\boldsymbol{z}_\epsilon$ to random noise.

$$q_{\phi_y}(y|\boldsymbol{z}_c, \boldsymbol{z}_d) = \text{Bernoulli}(y|\pi_{\phi_y}([\boldsymbol{z}_c; \boldsymbol{z}_d]), \quad q_{\sigma_\epsilon}(\boldsymbol{z}_\epsilon) = p_{\sigma_\epsilon}(\boldsymbol{z}_\epsilon), \tag{7}$$

where for distributions in eq. (6): $\{\boldsymbol{\mu}_{\phi_c}(.), \boldsymbol{\sigma}_{\phi_c}(.)\}$ is a neural network with parameters $\phi_c$; similarly for $\{\boldsymbol{\mu}_{\phi_d}(.), \boldsymbol{\sigma}_{\phi_d}(.)\}$ with $\phi_d$. We have $\boldsymbol{\sigma}_{\phi_c}(\boldsymbol{x})$ and $\boldsymbol{\sigma}_{\phi_d}(\boldsymbol{x})$ being diagonal covariance matrices. The distribution for random noise $\boldsymbol{z}_\epsilon$ is the same as in the generative model. In eq. (7), the $\pi_{\phi_y}(.)$ parametrizes the Bernoulli distribution $q_{\phi_y}(y|\boldsymbol{z}_c, \boldsymbol{z}_d)$, and is a neural network with parameters $\phi_y$. We found that using $q_{\phi_y}(y|\boldsymbol{z}_c, \boldsymbol{z}_d)$ yielded better results than using $q_{\phi_y}(y|\boldsymbol{z}_d)$ or $q_{\phi_y}(y|\boldsymbol{x})$. Since $\boldsymbol{z}_c$ provides important context on the clean patterns present, which in turn allows $\boldsymbol{z}_d$ to better focus on modelling dirty patterns. In practice, to stabilize the optimization procedure in a few cases, we used $\pi_{\phi_y}([\text{sg}(\boldsymbol{z}_c); \boldsymbol{z}_d])$ in $q_{\phi_y}(y|\boldsymbol{z}_c, \boldsymbol{z}_d)$ – eq. (7). Note that $\text{sg}(\boldsymbol{z}_c)$ stands for *stop gradient* operator applied to $\boldsymbol{z}_c$; and this prevents $\boldsymbol{z}_c$ from being updated with dirty pattern information early in the training. A more detailed discussion on $q_{\phi_y}(y|\boldsymbol{z}_c, \boldsymbol{z}_d)$ is presented in Annex E.

### 4.3 TRAINING LOSS

Our model is trained to maximize an objective function with three terms, which accounts for our semi-supervised setting. The first term $\mathcal{L}(\boldsymbol{x})$ is the evidence lower bound (ELBO) for the unlabelled part of the data. The second term $\mathcal{L}(\boldsymbol{x}, y)$ is the ELBO for the trusted set. The third term $\mathcal{L}_{\text{WCE}}(\boldsymbol{x}, y)$ is the weighted cross-entropy loss which ensures that $q_{\phi_y}(y|\boldsymbol{z}_c, \boldsymbol{z}_d)$ correctly predicts the trusted set labels $y$.

The ELBO for the unlabelled (unsupervised) part is

$$\begin{aligned} \mathcal{L}(\boldsymbol{x}) =&\, \mathbb{E}_{q_{\phi_c}(\boldsymbol{z}_c|\boldsymbol{x})q_{\phi_d}(\boldsymbol{z}_d|\boldsymbol{x})p_{\sigma_\epsilon}(\boldsymbol{z}_\epsilon)} \Big[ \pi_{\phi_y}([\boldsymbol{z}_c; \boldsymbol{z}_d]) \log p_\theta(\boldsymbol{x}|[\boldsymbol{z}_c; \boldsymbol{z}_\epsilon]) \\ &+ (1 - \pi_{\phi_y}([\boldsymbol{z}_c; \boldsymbol{z}_d])) \log p_\theta(\boldsymbol{x}|[\boldsymbol{z}_c; \boldsymbol{z}_d]) - D_{KL}\left(q_{\phi_y}(y|\boldsymbol{z}_c, \boldsymbol{z}_d)||p_\alpha(y)\right) \Big] \\ &- D_{KL}\left(q_{\phi_c}(\boldsymbol{z}_c|\boldsymbol{x})||p_{\sigma_c}(\boldsymbol{z}_c)\right) - D_{KL}\left(q_{\phi_d}(\boldsymbol{z}_d|\boldsymbol{x})||p_{\sigma_d}(\boldsymbol{z}_d)\right), \end{aligned} \tag{8}$$

where $q_{\sigma_\epsilon}(\boldsymbol{z}_\epsilon) = p_{\sigma_\epsilon}(\boldsymbol{z}_\epsilon)$ and so they cancel each other. The expectations are obtained via Monte-Carlo (MC) estimation via reparameterization trick (Kingma & Welling, 2014; Rezende et al., 2014).

For the trusted set (supervised) part the ELBO is

$$\begin{aligned} \mathcal{L}(\boldsymbol{x}, y) =&\, \mathbb{E}_{q_{\phi_c}(\boldsymbol{z}_c|\boldsymbol{x})q_{\phi_d}(\boldsymbol{z}_d|\boldsymbol{x})p_{\sigma_\epsilon}(\boldsymbol{z}_\epsilon)} \Big[ y \log p_\theta(\boldsymbol{x}|[\boldsymbol{z}_c; \boldsymbol{z}_\epsilon]) + (1 - y) \log p_\theta(\boldsymbol{x}|[\boldsymbol{z}_c; \boldsymbol{z}_d]) \Big] \\ &+ \log p_\alpha(y) - D_{KL}\left(q_{\phi_c}(\boldsymbol{z}_c|\boldsymbol{x})||p_{\sigma_c}(\boldsymbol{z}_c)\right) - D_{KL}\left(q_{\phi_d}(\boldsymbol{z}_d|\boldsymbol{x})||p_{\sigma_d}(\boldsymbol{z}_d)\right). \end{aligned} \tag{9}$$

Lastly, we need to define the *weighted cross-entropy* $\mathcal{L}_{\text{WCE}}(\boldsymbol{x}, y)$, which is

$$\mathcal{L}_{\text{WCE}}(\boldsymbol{x}, y) = -y.\log q(y = 1|\boldsymbol{x}) - \omega_{imb}.(1 - y).\log\left(1 - q(y = 1|\boldsymbol{x})\right) \tag{10}$$

where

$$\omega_{imb} = \max\left\{1, \frac{N_{l_1}}{N_{l_0}}\right\}, \quad N_{l_1} = \sum_{i=1}^{N_l} y_i, \quad N_{l_0} = N_l - N_{l_1}, \tag{11}$$

and $\omega_{imb}$ compensates for trusted set class imbalance.[2] However, we do not have access to $q(y = 1|\boldsymbol{x})$ and thus we cannot estimate $\mathcal{L}_{\text{WCE}}(\boldsymbol{x}, y)$ directly. Still, we can minimize an upper-bound

$$\begin{aligned} \mathcal{L}_{\text{WCE}}(\boldsymbol{x}, y) \le \tilde{\mathcal{L}}_{\text{WCE}}(\boldsymbol{x}, y) =&\, \mathbb{E}_{q_{\phi_c}(\boldsymbol{z}_c|\boldsymbol{x})q_{\phi_d}(\boldsymbol{z}_d|\boldsymbol{x})} \Big[ -y \log q(y = 1|\boldsymbol{z}_c, \boldsymbol{z}_d) \\ &- \omega_{imb}.(1 - y).\log\left(1 - q(y = 1|\boldsymbol{z}_c, \boldsymbol{z}_d)\right) \Big], \end{aligned} \tag{12}$$

which is obtained by applying Jensen's inequality.

Combining the three terms defined above, we *minimize* the overall loss

$$\mathcal{I} = -\frac{1}{N}\left[ \sum_{\boldsymbol{x} \in \mathcal{X}_u} \mathcal{L}(\boldsymbol{x}) + \sum_{(\boldsymbol{x}, y) \in \mathcal{X}_l \times \mathcal{Y}_l} \mathcal{L}(\boldsymbol{x}, y) \right] + \beta \frac{1}{N_l} \sum_{(\boldsymbol{x}, y) \in \mathcal{X}_l \times \mathcal{Y}_l} \tilde{\mathcal{L}}_{\text{WCE}}(\boldsymbol{x}, y) \tag{13}$$

with respect to the generative and variational parameters. The hyperparameter $\beta$ value controls the amount of up-sampling and importance relative to the other terms, which tends to be moderately high due to how small the trusted set is.

---

[2]Useful when the number of labelled outliers outnumbers the inliers. Such a case does not reflect the common dataset composition, i.e. the number of inliers is larger than outliers.

### 4.4 Distance Correlation Penalty

Ideally, $z_c$ captures clean patterns only, whilst $z_d$ captures dirty patterns. Therefore $z_c$ and $z_d$ should have low *mutual information* (MI). However, in more challenging scenarios, e.g. small trusted set or higher dataset corruption, obtaining this solution may not be guaranteed. Enforcing a constraint encouraging low MI between $z_c$ and $z_d$ will lead to better model performance and stability in challenging scenarios, improving repair quality.

We would like to introduce a constraint that minimizes MI between $z_c$ and $z_d$. However, approximating MI properly can be complex. Instead, we use *distance correlation* (DC) as a surrogate for MI (Székely et al., 2007), which is easier to compute and can measure non-linear dependencies between vector variables. Other works have used DC as a surrogate for MI, e.g. (Chen et al., 2021). Further, DC can also measure dependence between vector variables of different dimensions, which is often the case with $z_c$ and $z_d$. For the data batch $(z_c, z_d) \in (Z_c, Z_d)$ where $Z_c \in \mathbb{R}^{N \times q}$ and $Z_d \in \mathbb{R}^{N \times p}$, we can define the empirical estimate of DC as $dCorr_N(Z_c, Z_d)$. The definition of the estimator $dCorr_N(Z_c, Z_d)$ can be seen in Annex F. Essentially, DC is the standard correlation between the elements of the *double centered* pairwise distance matrices of each data batch $Z_c$ and $Z_d$. The range is $0 \leq dCorr_N(Z_c, Z_d) \leq 1$, where 0 means variables are independent, and 1 implies that $z_c$ and $z_d$ are strongly correlated.

Enforcing this constraint means adding a penalty to the model training loss. Hence, reusing the loss defined in eq. (13) we now have

$$\min_{\phi_c, \phi_d, \phi_y, \theta} \mathcal{I} + \lambda_t \, dCorr_N(Z_c, Z_d), \tag{14}$$

where $\lambda_t$ increases every epoch from 0 until it reaches a maximum value $\lambda_T$, which is then maintained. The rate of increase of $\lambda_t$ and $\lambda_T$ are hyper-parameters. This strategy is a type of penalty method as used in constrained optimization.

### 4.5 Outlier Detection and Repair Process

After training, we proceed with *outlier detection* and *automated repair*, as in Section 3. The detection task is to discover the ground-truth labels $y$ for each $x \in \mathcal{X}$, using inferred labels $\hat{y}$. A score $\mathcal{A}(x)$ and threshold $\gamma \geq 0$ are used to get the set of outliers $\mathcal{O} = \{x \in \mathcal{X} | \, \mathcal{A}(x) \geq \gamma\}$, where a higher $\mathcal{A}(x)$ means the more likely $x$ is an outlier. The inferred label $\hat{y}$ is obtained as: *(inlier)* $\hat{y} = 1$ if $x \notin \mathcal{O}$ ; *(outlier)* $\hat{y} = 0$ if $x \in \mathcal{O}$. For our model, we use a score based on the negative log probability of inlier given the latent subspaces, which is

$$\mathcal{A}(x) = -\log \pi_{\phi_y}([\boldsymbol{\mu}_{\phi_c}(x); \boldsymbol{\mu}_{\phi_d}(x)]), \quad x \in \mathcal{X}. \tag{15}$$

The threshold $\gamma$ can be chosen as $\gamma \approx -\log(0.5)$ assuming $q_{\phi_y}(y = 1|.) = \pi_{\phi_y}(.)$ is near calibrated, or it can be user-defined. The repair task is to obtain an inferred reconstruction $\hat{x}$ from the outlier $x \in \mathcal{O}$ such that it is close to the inlier ground-truth $\tilde{x}$. The repair is generated using the most likely reconstruction under our model for $y = 1$ (inliers), which means only the clean subspace $z_c$ is used. This is the *maximum a posteriori* estimate for a VAE, where one approximates $p_\theta(z_c|x)$ by $q_\phi(z_c|x)$, and then uses the means of $q_\phi(z_c|x)$, $p_\epsilon(z_\epsilon)$ and $p_\theta(x \, | \, [z_c; \, z_\epsilon])$ in the estimate. Hence, we have

$$\hat{x} = \boldsymbol{\mu}_\theta([\boldsymbol{\mu}_{\phi_c}(x); \mathbf{0}]), \quad x \in \mathcal{O}. \tag{16}$$

## 5 Experiments

We evaluate two tasks: *outlier detection*, and *automated repair*. Our experiments use three image datasets: *Frey-Faces*[3], *Fashion-MNIST* (Xiao et al., 2017), *Synthetic-Shapes*. *Synthetic-Shapes* is a synthetic dataset built around four different shapes (classes): a circle, a rectangle, an ellipse and a triangle. These are colored white and set in a black background. We corrupt datasets with synthetic systematic errors, since *public real-world* datasets with ground-truth repairs and respective labels are difficult to find, as seen in Eduardo et al. (2020); Krishnan et al. (2016); Liu et al. (2020). We compare our model (CLSVAE) with baselines ranging from completely supervised to unsupervised.[4]

---

[3]http://www.cs.nyu.edu/~roweis/data/frey_rawface.mat

[4]Code for models and experiments will be released upon paper acceptance.

| Dataset | Data Type | No. Data Classes | No. Error Classes | Error Types |
|---|---|---|---|---|
| *Synthetic-Shapes* | $28 \times 28$ binary (black / white) | 4 | 4 | 4 lines |
| *Fashion-MNIST* | $28 \times 28$ continuous (grey-scale) | 10 | 8 | 4 lines & 4 squares |
| *Frey-Faces* | $28 \times 20$ continuous (grey-scale) | 1 | 4 | 4 squares |

Table 1: Description of dataset and its corruption.

**Evaluation** For outlier detection we use *AVPR (Average Precision)* (Everingham et al., 2015) to measure detection quality, which is a surrogate for the area under the precision-recall curve (Hendrycks & Gimpel, 2016). This metric is preferred since it is insensitive to label imbalance, typical in outlier detection. AVPR score is between $[0, 1]$ and higher means better. For *automated repair* we want to quantify the quality of the repair, for outlier instances. We report the standardized mean squared error (SMSE) between pixels of the ground-truth (inlier) instance and that of the proposed repair. We report SMSE separately for the dirty pixels (those affected by the systematic error) and for the clean pixels (those unaffected). The first measures repair performance, while the latter measures *distortion* that the repair process causes to clean pixels. In both cases, a lower SMSE means better. Note in the case of binary pixels, the SMSE is just the Brier score, and thus is in $[0, 1]$.

**Datasets and Corruption Process** In our experiments we take an uncorrupted dataset and inject it with systematic errors. These systematic errors are synthetic, designed to seem like reasonable image corruptions, e.g. occlusion or failing of a camera sensor. The types of systematic errors used across datasets are either *lines* or *squares*. Lines (two diagonal, one vertical, and one horizontal) cross the image from side to side, and may have their color set at random (black / white). These lines always affect the same pixels, and have thickness of one pixel. Squares are randomly uniformly placed, so is their fill-in color, with size $6 \times 6$ pixels. We use different noise levels so we can study their impact, we use $[15\%, 25\%, 35\%, 45\%]$ of dataset. The systematic error corruption process is done by picking uniformly at random an instance, and then applying that systematic error. We use a range of trusted set sizes by defining the number of instances labelled per systematic error class (type) and per data class. The latter is the underlying classes in the dataset, e.g. item labels in *Fashion-MNIST*. For each class, either systematic error or data class, we provide label $y$ for a few instances at random obtaining a trusted set. We use the range $\text{TS}_{\text{size}} = [5, 10, 25, 50]$ labelled samples per class, which results in different trusted set sizes depending on number of classes. Particularly, we have the trusted set ranges: *Synthetic-Shapes* with $[40, 80, 200, 400]$ total samples; *Frey-Faces* with $[25, 50, 125, 250]$ total samples; *Fashion-MNIST* with $[90, 180, 450, 900]$ total samples. We create five examples (different random seeds) per noise level and per trusted set size, and train the models on them. The results are then averaged. Table 1 describes datasets and their corruption (error types), number of systematic errors and data classes. More details on this experimental setup in Annex G.

**Comparative Methods** Our baselines are VAEs since most of the relevant work in *sparse semi-supervision* is of this type (Ilse et al., 2020; Joy et al., 2020), and the task of repair is related with that of manipulating the reconstruction. We have four baselines: VAE-L2, CVAE, VAEGMM, CC-VAE. For details on model architecture and hyperparameters see Annex H. The VAE-L2 model is an unsupervised method tackling the issue of corruption by applying strong regularization ($\ell_2$ regularization on weights). VAE-L2 uses the reconstruction likelihood for detection, more details in Annex A. CVAE is the supervised version of the semi-supervised M2 model (Kingma et al., 2014), and should have better repair quality than M2. CVAE uses the reconstruction likelihood as an detection score. We found a smaller variance for $p(z)$ to be beneficial, for details see Annex B. VAEGMM, based on (Willetts et al., 2020), is an improved version of M2 for the sparse semi-supervision setting. In this setting the M2 model tends to have posterior collapse issues with $q(y|x)$, picking one class over others. VAEGMM overcomes this issue, improving clustering and classification performance. Hence, we expect competitive detection performance from VAEGMM, for details see Annex C. The CCVAE (Joy et al., 2020) is a state-of-the-art (SotA) semi-supervised disentanglement model, allowing attribute manipulation in semi-supervised settings. For repair, we follow the automatic attribute manipulation procedure proposed by (Joy et al., 2020) (see Annex D). We adapted their code to our pipeline. Contrary to Joy et al. (2020), we found performance was superior when using a large up-sampling coefficient for the classifier, i.e. like $\beta$ in CLSVAE. We provide two versions of our model, with and without distance correlation penalty in Section 4.4. So for CLSVAE-NODC use eq. (13) as training loss, whilst for CLSVAE use eq. (14).

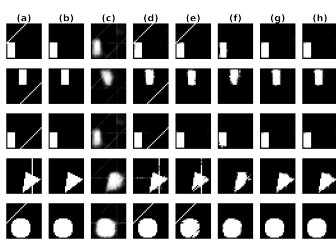
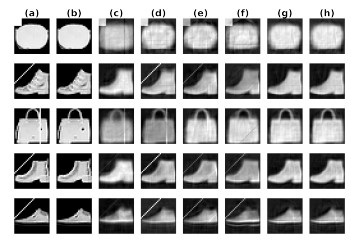
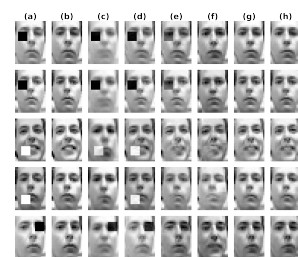

(a) *Synthetic-Shapes*: 35% noise, 10 labels per class (1.6% of dataset).

(b) *Fashion-MNIST*: 35% noise, 10 labels per class (0.25% of dataset).

(c) *Frey-Faces*: 35% noise, 10 labels per class (2.5% of dataset)

Figure 1: Images for model repair (reconstruction), outlier (corrupted) and inlier (uncorrupted): (a) Original (Outlier); (b) Ground-Truth (Inlier); (c) VAE-L2; (d) VAEGMM; (e) CVAE; (f) CCVAE; (g) CLSVAE-NODC; (h) CLSVAE. A larger version of this figure is found in Annex I.

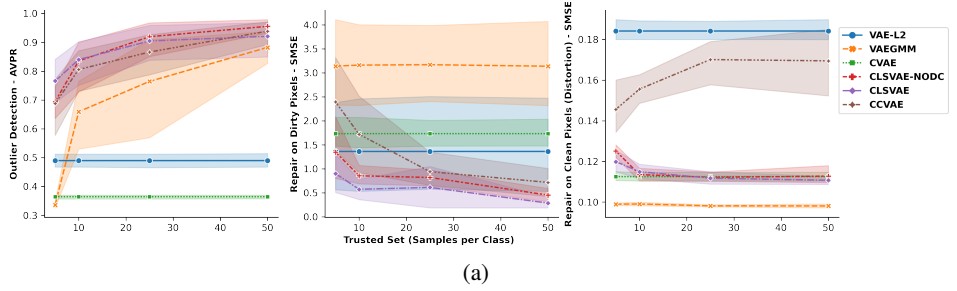

(a)

| Dataset | Model | Outlier Detection (AVPR ↑) | Repair on Dirty Pixels (SMSE ↓) | Repair on Clean Pixels (SMSE ↓) |
|---|---|---|---|---|
| *Synthetic-Shapes* | VAE-L2 (Kingma & Welling, 2014) | 0.93 (0.03) | 0.049 (0.008) | 0.015 (0.002) |
| | VAEGMM (Willetts et al., 2020) | 0.62 (0.10) | 0.974 (0.009) | 0.003 (3e-4) |
| | CVAE (Kingma et al., 2014) | 0.47 (0.03) | 0.429 (0.114) | 0.003 (3e-4) |
| | CCVAE (Joy et al., 2020) | **0.98** (0.03) | **0.031** (0.023) | 0.008 (0.001) |
| | CLSVAE-NODC (**Ours**) | **0.99** (3e-4) | **0.018** (0.024) | **0.002** (3e-4) |
| | CLSVAE (**Ours**) | **0.99** (0.02) | **0.014** (0.008) | **0.005** (0.004) |
| *Fashion-MNIST* | VAE-L2 (Kingma & Welling, 2014) | 0.49 (0.03) | **1.362** (1.140) | 0.175 (0.004) |
| | VAEGMM (Willetts et al., 2020) | **0.66** (0.13) | 3.161 (1.032) | **0.095** (0.001) |
| | CVAE (Kingma et al., 2014) | 0.36 (0.01) | 1.732 (0.335) | **0.099** (0.001) |
| | CCVAE (Joy et al., 2020) | **0.81** (0.09) | 1.719 (0.956) | 0.136 (0.003) |
| | CLSVAE-NODC (**Ours**) | **0.84** (0.10) | **0.854** (0.214) | 0.107 (0.002) |
| | CLSVAE (**Ours**) | **0.84** (0.08) | **0.572** (0.238) | 0.108 (0.002) |
| *Frey-Faces* | VAE-L2 (Kingma & Welling, 2014) | 0.73 (0.14) | 10.32 (6.118) | 0.420 (0.033) |
| | VAEGMM (Willetts et al., 2020) | **0.96** (0.06) | 22.32 (4.112) | **0.070** (0.003) |
| | CVAE (Kingma et al., 2014) | 0.42 (0.02) | 3.190 (0.675) | 0.111 (0.024) |
| | CCVAE (Joy et al., 2020) | **0.99** (0.01) | 0.947 (0.123) | 0.270 (0.059) |
| | CLSVAE-NODC (**Ours**) | **0.85** (0.13) | **0.269** (0.078) | 0.172 (0.048) |
| | CLSVAE (**Ours**) | **0.99** (0.02) | **0.321** (0.168) | 0.177 (0.033) |

(b)

Figure 2: Outlier detection uses AVPR score where **highest is best**. Repair for dirty pixels, and for clean pixels (distortion), uses SMSE where **lowest is best**. (a) Trusted set range sweep for *Fashion-MNIST* where [0.12, 0.25, 0.64, 1.28] % of the dataset, at 35 % noise level. (b) Table for results at 35% noise level, and 10 labelled samples per class for the trusted set. Boldface corresponds to the best performances within a standard error, and green color to best mean performance overall. Standard error in brackets.

## 5.1 DISCUSSION OF RESULTS

In Figure 2(a) we show performance as a function of the size of the trusted set, i.e. *sweep* of trusted set sizes, for a 35% noise level. Similar performance is seen for other datasets (see Annex J). Table 2(b) shows the results for all datasets for a 35% noise level and trusted set size of 10 labelled samples per class. Results for all trusted set sizes and noise levels are found in Annex J, with similar analysis on performance. Figure 1 shows some examples of image repairs for all datasets (larger version Annex I). Additional examples of repairs are seen in Annex K, including inlier instances, 45% noise level, and 5 samples per class (trusted set size) for *Synthetic-Shapes* since its an easier dataset.

**Outlier Detection** Looking at Figure 2, we see that on average both CLSVAE (our model) and CLSVAE-NODC have the highest AVPR, registering the best detection performance, with similar scores. We see that CCVAE, the previous SotA, has similar detection performance as CLSVAE.

VAEGMM, also semi-supervised, lags behind both likely because its designed for slightly larger trusted sets. All semi-supervised models (CLSVAE, CLSVAE-NODC, CCVAE, VAEGMM) improve their detection performance as the trusted set grows larger. CVAE and VAE-L2 do not use a trusted set, and thus have the same performance throughout all the trusted set range in Figure 2(a). These two observations about the trusted set range are also seen, for all datasets and noise levels, in Annex J. CVAE is supervised, still it shows poor performance, this may be due to: issues linked to (decoder) likelihood-based scores (Eduardo et al., 2020; Lan & Dinh, 2020); poor fitting to the data, thus impacting negatively the score. VAE-L2 uses a likelihood score, and is unsupervised, so poorer detection performance is understandable. This highlights semi-supervision as being important in systematic error detection. VAE-L2 registering good performance in *Synthetic-Shapes* (see Figure 2(b)) is likely due to this dataset being easier. Lastly, we note that CLSVAE tends to have better detection performance in higher noise levels relative to other methods (see Annex J). Therein, CCVAE has close to or similar detection performance as CLSVAE.

**Automated Repair** In Figure 2, we see that on average CLSVAE (our model) is best at automated repair (lowest SMSE on dirty pixels). We also see that distortion (repair clean pixels, SMSE) is relatively low, but not the lowest. This results in CLSVAE overall being the best repair method, not only replacing pixel values of the systematic error, but also inferring correctly the structure of the ground-truth repair (both clean and dirty pixels). This is confirmed in Figure 1, where reconstructions (repairs) by CLSVAE show the best quality: replacing the error values of affected pixels and recovering the underlying ground-truth, whilst preserving the uncorrupted image portion (i.e. low distortion). CLSVAE has slightly better repair than CLSVAE-NODC on average, but most importantly, it has better performance stability than CLSVAE-NODC, which can be seen in Annex J for repair (dirty pixels). Further, in Annex J, repair with CLSVAE is more advantageous relative to other models at higher noise levels. As expected, semi-supervised models (CLSVAE, CLSVAE-NODC, CCVAE) improve their repair of dirty pixels as the trusted set increases (see Figure 2(a)). Both VAE-L2 and CVAE do not use a trusted set, so performance is static. CCVAE has the ability to perform good repair, registering the second best repair for dirty pixels after CLSVAE, but often with higher distortion. CCVAE suffers from two issues that account for its worse performance relative to CLSVAE. For one, looking at Figure 1, CCVAE can sometimes fail to replace the pixel values from systematic errors. Secondly, and more often, it can fail to recover the underlying ground-truth even when replacing erroneous pixel values. Similarly, it has difficulty preserving the uncorrupted image portion (higher distortion). So some information about inlier appearance is being lost. This is explained by the fact that CCVAE latent space is not disentangled regarding the clean and dirty patterns. CVAE can repair some outliers well, but it fails to deal with other systematic errors, which leads to an overall poor repair performance. This is maybe due to the binary latent variable used for $y$ making it harder to model multiple systematic errors, for more discussion see Joy et al. (2020). VAE-L2 is able to repair some errors, but overall has worse repair than CLSVAE. Its strong regularization, optimized mostly for detection, leads to higher distortion and loss of detail (see Figure 1). Its higher standard error (erratic repairs) is due to not being able to distinguish between clean and dirty patterns. VAEGMM does not do well in repair. This is likely due to it being better suited for classification or clustering tasks.

## 6 CONCLUSION

We have proposed a novel semi-supervised VAE (CLSVAE) for outlier detection and automated repair, in the presence of systematic errors. Our model exploits the fact that systematic errors are predictable by high capacity models, unlike random errors. Thus, CLSVAE partitions the latent space into two subspaces: one for clean patterns, and another for dirty ones. Inliers are only modelled by the clean pattern subspace, whilst outliers use both subspaces. We encourage low mutual information between these subspaces through a penalty, improving performance stability. Empirically this encourages higher fidelity repairs by the model, without human in the loop or other post-processing. We show CLSVAE only needs a small trusted set, requiring the user to label less data. We show that unsupervised models may not be able to distinguish between clean patterns and systematic errors, and strong regularization leads to a lower quality repair. Experimentally, CLSVAE showed superior repair quality and performance compared to other semi-supervised models, including a SotA disentanglement model. Experiments were carried out on image data, and in the future, we would like to explore other types of systematic errors and data types (tabular, sensor, natural language).

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

## A  VAE-$L_2$ (UNSUPERVISED)

The unsupervised VAE can be used to perform both outlier detection and repair. However, without regularization it will overfit to systematic errors (outliers). So, here we regularize both the encoder and decoder weights via $\ell_2$ regularization (weight decay). We can write the modified ELBO as

$$\log p(\boldsymbol{x}) \geq \mathcal{L}(\boldsymbol{x}) = \mathbb{E}_{q_\phi(\boldsymbol{z}|\boldsymbol{x})}\left[p_\theta(\boldsymbol{x}|\boldsymbol{z})\right] - D_{KL}\left(q_\phi(\boldsymbol{z}|\boldsymbol{x})||p(\boldsymbol{z})\right) + \lambda_{\ell_2} \sum_{w_i \in \mathcal{W}_{\text{AE}}} ||w_i||^2, \quad (17)$$

where $\theta$ and $\phi$ are the parameters for the decoder and encoder respectively. Further, $\lambda_{\ell_2}$ defines the regularization strength. The $w_i$ are the parameters for a layer indexed by $i$, belonging to the set of all VAE layer parameters $\mathcal{W}_{\text{AE}}$. Note that $\lambda_{\ell_2}$ is picked by using trusted set performance in outlier detection (AVPR) and repair (SMSE) metrics. Using repair metrics is not possible in practice, under our problem formulation, since we only have labels about which instances are outliers / inliers. However we relaxed this assumption here, since we wanted to obtain the best results possible with VAE-L2.

In terms of outlier detection, we used an *anomaly score* based on the reconstruction (negative log-likelihood) of the VAE. This type of anomaly score is typical in VAEs. For each $d \in \{0, ..., D\}$ features, we have

$$\mathcal{A}(\boldsymbol{x}) = -\sum_d^D \log p_\theta(x_d|\boldsymbol{\mu}_\phi(\boldsymbol{x})), \quad \boldsymbol{x} \in \mathcal{X}, \quad (18)$$

where the outlier set of points is $\mathcal{O} = \{\boldsymbol{x} \in \mathcal{X}| \ \mathcal{A}(\boldsymbol{x}) \geq \gamma\}$, where $\gamma$ is decided by the user. The repair process for the outliers is

$$\hat{\boldsymbol{x}} = \boldsymbol{\mu}_\theta(\boldsymbol{\mu}_\phi(\boldsymbol{x})), \quad \boldsymbol{x} \in \mathcal{O}. \quad (19)$$

## B  CVAE (FULLY SUPERVISED)

The CVAE *(Conditional VAE)* model (Kingma et al., 2014; Sohn et al., 2015) needs the entire train set to be labelled, i.e. like an *oracle* where the ground-truth for $y$ is observed. This is generally not possible in practice, particularly for our model, as it would be *cheating* under the problem formulation. But it does provide a valuable baseline about what kind of performance this type of model, given all advantages, can obtain. A semi-supervised version of this model is in fact the M2 model (Kingma et al., 2014), which will always register lower performance than its supervised counter-part the CVAE. The ELBO for this model is

$$\log p(\boldsymbol{x}, y) \geq \mathcal{L}(\boldsymbol{x}, y) = \mathbb{E}_{q_\phi(\boldsymbol{z}|\boldsymbol{x}, y)}\left[p_\theta(\boldsymbol{x}|\boldsymbol{z}, y)\right] - D_{KL}\left(q_\phi(\boldsymbol{z}|\boldsymbol{x}, y)||p_\sigma(\boldsymbol{z})\right), \quad (20)$$

where $\theta$ and $\phi$ are the decoder and encoder parameters. Further, note that in a standard CVAE we usually have $p(\boldsymbol{z}) = \mathcal{N}(\boldsymbol{z}|\boldsymbol{0}, \boldsymbol{I})$. However, we found empirically that stronger regularization for the encoder was needed for CVAE to do well in repair. Otherwise, it would often overfit to the systematic errors. Hence we used a modified prior $p_\sigma(\boldsymbol{z}) = \mathcal{N}(\boldsymbol{z}|\boldsymbol{0}, \sigma^2\boldsymbol{I})$ where $\sigma \in [0.1, 0.8]$, thus enforcing stronger regularization *(lower message capacity)* on latent code $\boldsymbol{z}$. Even though it is not standard, we also tried $q_\phi(\boldsymbol{z}|\boldsymbol{x})$ but always ended up with a bit worse performance than $q_\phi(\boldsymbol{z}|\boldsymbol{x}, y)$.

In terms of outlier detection, we used an *anomaly score* based on the reconstruction (negative log-likelihood) of the CVAE model. This type of anomaly score is typical in VAE type models. For each

$d \in \{0, ..., D\}$ features, we have

$$\mathcal{A}(\boldsymbol{x}) = -\sum_{d}^{D} \log p_\theta(x_d | \boldsymbol{\mu}_\phi(\boldsymbol{x}), y = 1), \quad \boldsymbol{x} \in \mathcal{X}, \tag{21}$$

where the outlier set of points is $\mathcal{O} = \{\boldsymbol{x} \in \mathcal{X} | \mathcal{A}(\boldsymbol{x}) \geq \gamma\}$, where $\gamma$ is decided by the user. We followed the CVAE attribute manipulation strategy from (Klys et al., 2018) when defining repair of an outlier. We encode using the original label (i.e. $y = 0$), then switch label in latent space (i.e. $y = 1$) and decode it. In other words, for repair we have

$$\hat{\boldsymbol{x}} = \boldsymbol{\mu}_\theta(\boldsymbol{\mu}_\phi(\boldsymbol{x}, y = 0), y = 1), \quad \boldsymbol{x} \in \mathcal{O}. \tag{22}$$

## C  VAEGMM: ALTERNATIVE TO M2 (SPARSE SEMI-SUPERVISED)

This formulation is based on Willetts et al. (2020), specifically GM-DGM (Gaussian Mixture Deep Generative Model). The work focused on severe sparse semi-supervision (some classes do not even have labels). This formulation is relevant, in sparse semi-supervision, since posterior collapse of $q(y|\boldsymbol{x})$ can occur for the original M2 model (Kingma et al., 2014). This in turn leads to poor performance in classification and clustering tasks. Hence, one can think of Willetts et al. (2020) as an improvement on M2 model for the issue above. We now present the version of GM-DGM from Willetts et al. (2020) applied to our problem.

### C.1  GENERATIVE MODEL

The generative model is defined as

$$p_\theta(\boldsymbol{x}|\boldsymbol{z})p_\tau(\boldsymbol{z}|y)p_\alpha(y), \tag{23}$$

where

$$p_\alpha(y) = \text{Bernoulli}(y|\alpha), \tag{24}$$

$$p_\tau(\boldsymbol{z}|y) = y\,\mathcal{N}(\boldsymbol{z}|\boldsymbol{0}, \sigma_{y=1}^2\boldsymbol{I}) + (1-y)\,\mathcal{N}(\boldsymbol{z}|\boldsymbol{0}, \sigma_{y=0}^2\boldsymbol{I}), \tag{25}$$

$$p_\theta(\boldsymbol{x}|\boldsymbol{z}) = \mathcal{N}(\boldsymbol{x}|\boldsymbol{\mu}_\theta(\boldsymbol{z}), \boldsymbol{\Sigma}_\theta(\boldsymbol{z})), \tag{26}$$

which defines a 2-component *Gaussian Mixture Model* w.r.t. $\boldsymbol{z}$, and $\tau = \{\sigma_{y=1}, \sigma_{y=0}\}$ and $\sigma_{y=1} < \sigma_{y=0}$. A sensible range for $\tau$ is: $\sigma_{y=1} = [0.2, 1]$; $\sigma_{y=1} = [2, 8]$. This prior defines a 2-component *Richter distribution* (Gales & Olsen, 1999), expressing a type of *heavy-tailed distribution* on $\boldsymbol{z}$. This type of distribution is common in robust statistics (Eduardo et al., 2020), as a means to robustify (regularize) the parameters of a model to outliers. The main idea here is that inlier samples $\boldsymbol{z}|y = 1$ will be regularized more strongly compared to outliers $\boldsymbol{z}|y = 0$, reflected by $\sigma_{y=1} < \sigma_{y=0}$. We initially *tried learning the parameters of the Gaussian components*. However, we obtained *much better results by fixing their parameters* (i.e. mean vector and covariance matrix), particularly when it came to outlier detection with smaller trusted sets. Lastly $\alpha$ reflects the initial belief on the fraction of clean data.

### C.2  VARIATIONAL MODEL

As for the variational model, we use the standard formulation provided in Willetts et al. (2020), which is the same as the original M2 model (Kingma et al., 2014). So we have

$$q_\phi(\boldsymbol{z}|\boldsymbol{x}, y)q_\phi(y|\boldsymbol{x}), \tag{27}$$

$$q_\phi(y|\boldsymbol{x}) = \text{Bernoulli}(y|\pi_\phi(\boldsymbol{x})), \tag{28}$$

$$q_\phi(\boldsymbol{z}|\boldsymbol{x}, y) = \mathcal{N}(\boldsymbol{z}|\boldsymbol{\mu}_\phi(\boldsymbol{x}, y), \boldsymbol{\Sigma}_\phi(\boldsymbol{x}, y)), \tag{29}$$

where $\pi_\phi(\boldsymbol{x})$, $\boldsymbol{\mu}_\phi(\boldsymbol{x}, y)$ and $\boldsymbol{\Sigma}_\phi(\boldsymbol{x}, y)$ are neural networks.

### C.3 TRAINING LOSS

We can write the ELBO (Evidence Lower Bound) for the unsupervised part of the dataset, i.e. $\boldsymbol{x} \in \mathcal{X}_u$, where labels $y$ are not observed as

$$\log p(\boldsymbol{x}) \geq \mathcal{L}(\boldsymbol{x}) = \int q_\phi(\boldsymbol{z}|\boldsymbol{x}, y) q_\phi(y|\boldsymbol{x}) \log \frac{p_\theta(\boldsymbol{x}|\boldsymbol{z}) p_\tau(\boldsymbol{z}|y) p_\alpha(y)}{q_\phi(\boldsymbol{z}|\boldsymbol{x}, y) q_\phi(y|\boldsymbol{x})} d\boldsymbol{z} dy \tag{30}$$

$$\mathcal{L}(\boldsymbol{x}) = \mathbb{E}_{q_\phi(y|\boldsymbol{x}) q_\phi(\boldsymbol{z}|\boldsymbol{x}, y)} \left[ \log p_\theta(\boldsymbol{x}|\boldsymbol{z}) \right] - \tag{31}$$
$$- \mathbb{E}_{q_\phi(y|\boldsymbol{x})} \left[ D_{KL}(q_\phi(\boldsymbol{z}|\boldsymbol{x}, y) || p_\tau(\boldsymbol{z}|y)) \right] - D_{KL}(q_\phi(y|\boldsymbol{x}) || p_\alpha(y)).$$

One can rewrite the above ELBO in a different way as

$$\mathcal{L}(\boldsymbol{x}) = \pi_{\phi(\boldsymbol{x})} \mathcal{G}(\boldsymbol{x}, y = 1) + \left( 1 - \pi_{\phi(\boldsymbol{x})} \right) \mathcal{G}(\boldsymbol{x}, y = 0) - D_{KL}(q_\phi(y|\boldsymbol{x}) || p_\alpha(y)), \tag{32}$$

where

$$\mathcal{G}(\boldsymbol{x}, y) = \mathbb{E}_{q_\phi(\boldsymbol{z}|\boldsymbol{x}, y)} \left[ \log p_\theta(\boldsymbol{x}|\boldsymbol{z}) \right] - D_{KL}(q_\phi(\boldsymbol{z}|\boldsymbol{x}, y) || p_\tau(\boldsymbol{z}|y)), \tag{33}$$

and $\pi_{\phi(\boldsymbol{x})} = q_\phi(y = 1|\boldsymbol{x})$ is the probability a data instance is clean.

The ELBO for the labelled part of the dataset (i.e. trusted set), where we have $(\boldsymbol{x}, y) \in \mathcal{X}_l, \mathcal{Y}_l$, is written as

$$\log p(\boldsymbol{x}, y) \geq \mathcal{L}(\boldsymbol{x}, y) = \mathcal{G}(\boldsymbol{x}, y) + \log p_\alpha(y). \tag{34}$$

Given the aforementioned definitions for the model and ELBOs (labelled and unlabelled), the overall training loss is the same as eq. (13) from the main paper.

### C.4 OUTLIER DETECTION AND REPAIR PROCESS

After training the model, then we can proceed with the *outlier detection* and *repair* process. First, we need to define a score for use in detection. So at test time we use

$$\mathcal{A}(\boldsymbol{x}) = -\log q_\phi(y = 1|\boldsymbol{x}), \quad \boldsymbol{x} \in \mathcal{X}, \tag{35}$$

which is based on the probability of the instance being an inlier. Usually, the user defines a threshold $\gamma$ to classify instances into inliers or outliers. The set of outlier instances is given by $\mathcal{O} = \{\boldsymbol{x} \in \mathcal{X} | \mathcal{A}(\boldsymbol{x}) \geq \gamma\}$, where $\gamma \geq 0$. Assuming $q_\phi(y = 1|\boldsymbol{x})$ is somewhat calibrated, then one can use a $\gamma \approx -\log(0.5)$. After obtaining the set $\mathcal{O}$, and conditioning on $y = 1$, we produce a repair using

$$\hat{\boldsymbol{x}} = \boldsymbol{\mu}_\theta(\boldsymbol{\mu}_\phi(\boldsymbol{x}, y = 1)), \quad \boldsymbol{x} \in \mathcal{O}. \tag{36}$$

## D CCVAE (SEMI-SUPERVISED DISENTANGLEMENT)

In Joy et al. (2020) the latent space $\boldsymbol{z}$ is split into two subspaces: the style (or agnostic) part $\boldsymbol{z}_{\backslash c}$ that is meant to model unlabelled patterns present in the instance; the characteristics part $\boldsymbol{z}_c$ that is meant to model the labelled attributes (characteristics). Formally, we have $\boldsymbol{z} = \left[ \boldsymbol{z}_c; \boldsymbol{z}_{\backslash c} \right]$ where $[ \, ; \, ]$ is the concatenation operation. In our case, following Joy et al. (2020) and their implementation code, we modelled $\boldsymbol{z}_c$ as single variable (univariate) for our binary label: $y = 1$ for inliers and $y = 0$ for outliers.

### D.1 GENERATIVE MODEL

From Joy et al. (2020), using a similar notation, we have

$$p(\boldsymbol{x}, \boldsymbol{z}, y) = p_\theta(\boldsymbol{x}|\boldsymbol{z}) p_\psi(\boldsymbol{z}_c|y) p(\boldsymbol{z}_{\backslash c}) p(y), \tag{37}$$

$$p(y) = \text{Bernoulli}(y|\alpha), \tag{38}$$

$$p(\boldsymbol{z}_{\backslash c}) = \mathcal{N}(\boldsymbol{z}_{\backslash c}|\boldsymbol{0}, \mathbf{I}), \tag{39}$$

$$p_\psi(\boldsymbol{z}_c|y) = \mathcal{N}(\boldsymbol{z}_c|\boldsymbol{\mu}_\psi(y), \boldsymbol{\sigma}_\psi^2(y)), \tag{40}$$

where $p_\theta(\boldsymbol{x}|\boldsymbol{z})$ is a neural network based decoder, and $\theta$ are the parameters of the neural network. In this case $\alpha$ has the same meaning as in CLSVAE, where it expresses the prior belief on the fraction of inliers present in the dataset. Note that the above model expresses a two-component mixture model on $\boldsymbol{z}_c$, with $y$ as gating variable. In fact we have a mean and variance for each $y$ value as it pertains to $\boldsymbol{z}_c$, i.e. $\boldsymbol{\mu}_\psi(y = 1)$ and $\boldsymbol{\sigma}_\psi(y = 1)$ for inliers and similar for outliers ($y = 0$).

## D.2 Variational Model

The variational distribution is factorized as

$$q(y, \boldsymbol{z}|\boldsymbol{x}) = q_{\varphi,\phi}(y|\boldsymbol{x}) \, q_{\varphi,\phi}(\boldsymbol{z}|\boldsymbol{x}, y), \tag{41}$$

$$q_{\varphi}(y|\boldsymbol{z}_c) = \text{Bernoulli}(y|\pi_{\varphi}(\boldsymbol{z}_c)), \tag{42}$$

$$q_{\phi}(\boldsymbol{z}|\boldsymbol{x}) = \mathcal{N}(\boldsymbol{z}|\boldsymbol{\mu}_{\phi}(\boldsymbol{x}), \boldsymbol{\sigma}_{\phi}^2(\boldsymbol{x})), \tag{43}$$

$$q_{\varphi,\phi}(y|\boldsymbol{x}) = \int q_{\varphi}(y|\boldsymbol{z}_c) q_{\phi}(\boldsymbol{z}|\boldsymbol{x}) d\boldsymbol{z}, \tag{44}$$

$$q_{\varphi,\phi}(\boldsymbol{z}|\boldsymbol{x}, y) = \frac{q_{\varphi}(y|\boldsymbol{z}_c) q_{\phi}(\boldsymbol{z}|\boldsymbol{x})}{q_{\varphi,\phi}(y|\boldsymbol{x})}, \tag{45}$$

where $q_{\varphi}(y|\boldsymbol{z}_c)$ and $q_{\phi}(\boldsymbol{z}|\boldsymbol{x})$ are neural network based encoders, with $\varphi$ and $\phi$ as neural network parameters.

## D.3 Training Loss

The training loss used in Joy et al. (2020) is defined by the ELBO, with one term for the labelled data (trusted set) and a second term the unlabelled data. The total ELBO is written as

$$\sum_{\boldsymbol{x} \in \mathcal{X}_u} \mathcal{L}_{\text{CCVAE}}(\boldsymbol{x}) + \sum_{(\boldsymbol{x}, y) \in \mathcal{X}_l \times \mathcal{Y}_l} \mathcal{L}_{\text{CCVAE}}(\boldsymbol{x}, y), \tag{46}$$

where the labelled part of the ELBO is

$$\mathcal{L}_{\text{CCVAE}}(\boldsymbol{x}, y) = \mathbb{E}_{q_{\phi}(\boldsymbol{z}|\boldsymbol{x})} \left[ \frac{q_{\varphi}(y|\boldsymbol{z}_c)}{q_{\varphi,\phi}(y|\boldsymbol{x})} \log \left( \frac{p_{\theta}(\boldsymbol{x}|\boldsymbol{z}) p_{\psi}(\boldsymbol{z}_c|y)}{q_{\varphi}(y|\boldsymbol{z}_c) q_{\phi}(\boldsymbol{z}|\boldsymbol{x})} \right) \right] + \beta \log q_{\varphi}(y|\boldsymbol{x}) + \log p(y), \tag{47}$$

where $\beta$ is the hyperparameter controlling amount of up-sampling and importance relative to other terms, like in CLSVAE. In Joy et al. (2020), for their application, they found that setting $\beta = 1$ brought good results and found no need to tune it further. In our case, we found that we obtained better performance by using larger values for $\beta$. This is probably because the application is different, i.e. outlier detection and subsequent repair, and since the trusted set (labelled set) is quite small. The unlabelled part of the ELBO is not as important for our analysis here, please see the original paper.

## D.4 Outlier Detection and Repair

After training the model, we proceed with outlier detection and automated repair. We need to define a score for use in detection, and for that we use the classifier given by the variational model. As such, we have

$$\mathcal{A}(\boldsymbol{x}) = -\log \mathbb{E}_{q_{\phi}(\boldsymbol{z}|\boldsymbol{x})} \left[ q_{\varphi}(y = 1|\boldsymbol{z}_c) \right], \quad \boldsymbol{x} \in \mathcal{X}, \tag{48}$$

which is the approximate negative log-probability of an instance being an inlier. The user may define a threshold $\gamma$ to classify instances into inliers or outliers. The set of outlier instances is given by $\mathcal{O} = \{\boldsymbol{x} \in \mathcal{X}| \, \mathcal{A}(\boldsymbol{x}) \geq \gamma\}$, where $\gamma \geq 0$. Assuming the classifier is somewhat calibrated, then one can use the a $\gamma \approx -\log(0.5)$.

From the perspective of CCVAE, the repair of an outlier instance is just attribute manipulation via $\boldsymbol{z}_c$ subspace. Once the appropriate $\boldsymbol{z}_c$ is found, then one uses the decoder for reconstruction obtaining a repair. Under our problem definition, automated repair is very much like *conditional generation* as seen in Joy et al. (2020), where samples for $\boldsymbol{z}_c$ are drawn from the conditional prior whilst reusing $\boldsymbol{z}_{\backslash c}$ obtained from encoding the outlier instance. Specifically, we have $\boldsymbol{z} = \left[ \boldsymbol{z}_c \sim p_{\psi}(\boldsymbol{z}_c|y); \boldsymbol{z}_{\backslash c} \right]$, where depending on the $y$ value it forces the generated samples to have, or not have, the presence of the attribute (e.g. inlier / outlier). In our case, we are interested in automated repair, and thus limiting human interaction apart from building the trusted set. That means that exploring $\boldsymbol{z}_c$ with user interaction to pick the best repair (reconstruction) for each outlier instance is not realistic. So we wish to obtain the most likely reconstruction under $y = 1$, i.e. the *maximum a posteriori*, thus defining an automated repair for the outlier instance. Hence, the repair is given by

$$\hat{\boldsymbol{x}} = \boldsymbol{\mu}_{\theta} \left( \left[ \boldsymbol{\mu}_{\psi}(y = 1); \boldsymbol{\mu}_{\phi}(\boldsymbol{x})_{\backslash c} \right] \right), \quad \boldsymbol{x} \in \mathcal{O}, \tag{49}$$

where $\boldsymbol{\mu}_{\psi}(y = 1)$ is the mean for the inlier component of $p_{\psi}(\boldsymbol{z}_c|y)$, and $\boldsymbol{\mu}_{\phi}(\boldsymbol{x})_{\backslash c}$ is the mean of $q_{\phi}(\boldsymbol{z}_{\backslash c}|\boldsymbol{x})$, which excludes the characteristic (labelled) latent subspace.

# E  DISCUSSION ABOUT CLASSIFIER IN CLSVAE

We found that using $q_{\phi_y}(y|z_c, z_d)$ leads to better performance in more challenging scenarios, e.g. smaller trusted set or when higher data corruption is present. One could have used $q_{\phi_y}(y|z_d)$, which works, but it is still worse than the former. This is mostly because $z_c$ provides important context on the clean patterns present in the instance, which in turn allows for $z_d$ to contain less information about clean patterns. Hence, $z_d$ will be able to focus more on the dirty patterns, and so the mutual information (MI) between $z_c$ and $z_d$ will be lower.

So the idea of using the *stop gradient* operator in $q_{\phi_y}(y|z_c, z_d) = \text{Bernoulli}(y|\pi_{\phi_y}([\text{sg}(z_c); z_d])$ has to do with preventing $z_c$ from having dirty pattern information. By using $\text{sg}(z_c)$, when executing back-propagation, we prevent the gradient flow from $q_{\phi_y}(y|z_c, z_d)$ to influence the learning of parameters $\phi_c$. As such, $\phi_c$ will only be affect by gradients stemming from the reconstruction of inlier instances, and not the decision on $y$. This trick is useful early on in the training, stabilizing the outcome of the optimization procedure, leading to better repair. Generally, only a few cases benefit from this trick, and usually when small trusted sets are used.

Further, $q_{\phi_y}(y|z_c, z_d)$ allows for the cross-entropy loss used for the trusted set to directly bias the latent space. This means labels $y \in \mathcal{Y}_l$ can help in separating the latent inlier and outlier representations (manifolds). Other works have tried biasing the latent space in the same fashion, e.g. (Locatello et al., 2019; Ilse et al., 2020; Joy et al., 2020).

# F  EMPIRICAL DISTANCE CORRELATION

For a random data batch of $n$ samples, such as $(\boldsymbol{Z}, \boldsymbol{Z}') = \{(z_k, z_k') : k = 1, ..., n\}$, one can define the empirical estimate for the *distance correlation* between multivariate random variables $z \in \mathbb{R}^q$ and $z' \in \mathbb{R}^p$. Note that $q$ and $p$ can be of different dimensions, i.e. $q \neq p$. Given this data batch, using the definition from (Székely et al., 2007), we first define $A_{kl}$ and $B_{kl}$ as

$$a_{kl} = ||z_k - z_l||_2 \ ,$$

$$\bar{a}_{k\cdot} = \frac{1}{n} \sum_{l=1}^{n} a_{kl} \ , \qquad \bar{a}_{\cdot l} = \frac{1}{n} \sum_{k=1}^{n} a_{kl} \ , \qquad \bar{a}_{\cdot\cdot} = \frac{1}{n^2} \sum_{k,l=1}^{n} a_{kl} \ ,$$

$$A_{kl} = a_{kl} - \bar{a}_{k\cdot} - \bar{a}_{\cdot l} + \bar{a}_{\cdot\cdot} \ ,$$

and similarly, using $b_{kl} = ||z_k' - z_l'||_2$ we define

$$B_{kl} = b_{kl} - \bar{b}_{k\cdot} - \bar{b}_{\cdot l} + \bar{b}_{\cdot\cdot} \ .$$

Now we are ready to define the empirical *distance covariance* as

$$dCov_n(\boldsymbol{Z}, \boldsymbol{Z}') = \frac{1}{n^2} \sum_{k,l=1}^{n} A_{kl} B_{kl} \ ,$$

and the following empirical *distance variances* for each random variable as

$$dVar_n(\boldsymbol{Z}) = \frac{1}{n^2} \sum_{k,l=1}^{n} A_{kl}^2 \ , \qquad dVar_n(\boldsymbol{Z}') = \frac{1}{n^2} \sum_{k,l=1}^{n} B_{kl}^2 \ .$$

Finally, combining the above measures we can write the *distance correlation* as

$$dCorr_n(\boldsymbol{Z}, \boldsymbol{Z}') == \frac{dCov_n(\boldsymbol{Z}, \boldsymbol{Z}')}{\sqrt{dVar_n(\boldsymbol{Z}) \ dVar_n(\boldsymbol{Z}')}} \ ,$$

where $0 \leq dCorr_n(\boldsymbol{Z}, \boldsymbol{Z}') \leq 1$. Note that if $dCorr_n(\boldsymbol{Z}, \boldsymbol{Z}') = 0$ then $z$ and $z'$ are statistically independent random variables. Otherwise, if $dCorr_n(\boldsymbol{Z}, \boldsymbol{Z}') = 1$ then they are strongly correlated. The *distance correlation* measure can capture non-linear dependencies, whilst the more common Pearson correlation can only capture linear dependencies. Further, a value of $0$ for Pearson correlation does not imply independence, unlike in *distance correlation* which it does.

## G  EXPERIMENTS: DETAILS ON DATASETS, CORRUPTION, AND TRUSTED SETS

Here we have a detailed description of the datasets and their corruption. For each noise level corruption in $\text{NL}_{\text{size}} = [15\%, 25\%, 35\%, 45\%]$, we instantiate five different examples of the same dataset using different random seeds. For each of those examples we build several trusted sets using the sizes in $\text{TS}_{\text{size}} = [5, 10, 25, 50]$, where labelled instances of smaller trusted sets are reused in bigger ones. The models are run on each example, for each trusted set, and results are then averaged.

**Synthetic-Shapes** This is a synthetic image dataset. It is meant to test the models in a simpler setting as it relates to the clean dataset. We treat the pixel values as a Bernoulli variable. The underlying clean dataset is composed of four different shapes (classes): a circle, a rectangle, an ellipse and a triangle. The shapes are filled by white pixels and the background is black. For each instance, the shape is placed uniformly at random inside the $28 \times 28$ black background. The systematic errors have four types, all are white lines that cross the square image from one side to another side. We have 4 fixed-in-place lines (two diagonal, one vertical, and one horizontal), affecting the same pixels, and sometimes intersecting with the shapes. Hence, we have a total of eight underlying classes for trusted set constitution, i.e. four data and four systematic error classes.

The size of the images is $28 \times 28$, and the pixel values are binary $\{0, 1\}$, i.e. black and white. Overall, we have a dataset size of $N = 5000$, with the following split: train (80%), validation (10%), test (10%). Given the eight underlying classes, the size dataset $N$, and $\text{TS}_{\text{size}}$, we have the trusted set range: $[40, 80, 200, 400]$ total number of labelled instances, which corresponds to $[0.8\%, 1.6\%, 4\%, 8\%]$ of the entire dataset

**Frey-Faces** This is a gray-scale image dataset consisting of the same person with different facial expressions. We treat pixel values as continuous. In terms of data classes, we only have one (monolithic), since no labels for the expressions are provided. We have four systematic error classes, which consist of four randomly uniformly placed squares of $6 \times 6$ pixels. The place and color of these squares is defined by the random seed, when the corruption example is created. After that, always the same features are affected. Hence, in total we have five underlying classes for trusted set constitution, i.e. one data and four systematic error classes.

The size of images is $28 \times 20$, and the pixel values range from $[0, 256]$, i.e. gray-scale. The size of the entire dataset is $N = 1965$, with the following split: train (80%), validation (10%), test (10%). Given the above and $\text{TS}_{\text{size}}$, the trusted set range is: $[25, 50, 125, 250]$ total number of labelled instances, which corresponds to $[1.3\%, 2.5\%, 6.4\%, 12.7\%]$ of the entire dataset.

**Fashion-MNIST** This is a gray-scale image dataset, which consists of images of different types of clothing and accessories from an online merchant. There are 10 existing data classes, provided with the dataset. We have 4 fixed-in-place lines (two diagonal, one vertical, and one horizontal) where the color (black or white) for each depends on the random seed. Then we have 4 squares of size $6 \times 6$ placed randomly uniformly with random color, dependent on random seed. Hence, we have a total of 18 underlying classes for trusted set constitution, i.e. 10 data and 8 systematic error classes.

The size of the images is $28 \times 28$, and the pixel values are continuous with range $[0, 1]$, i.e. gray-scale. The original train set is of size 60000 instances, which we split for our actual train set of 54000 (90%) and validation set of 6000 (10%). We use the same test set of 10000 instances, so $N = 70000$. Given $\text{TS}_{\text{size}}$ and the train set size $N$, the trusted set range is: $[90, 180, 450, 900]$ total number of labelled instances, which corresponds to $[0.12\%, 0.25\%, 0.64\%, 1.28\%]$ of the entire dataset.

## H  EXPERIMENTS: MODEL ARCHITECTURES, HYPER-PARAMETERS AND OPTIMIZATION

**Hyperparameter Selection**

In our experiments, we tuned model hyperparameters according to outlier detection performance, which means highest AVPR (average precision). This was evaluated on the trusted set, the only labelled part of the dataset. Often, we would look at the repairs (reconstructions) offered by each model for the trusted set. This way we confirm that the repair process is reasonable enough, and no adjustment is needed on the hyperparameters side. In the case of VAE-L2 we had to not just

account for AVPR in the trusted set, but also check repair performance via SMSE (standardized mean squared error). This is because strong regularization, higher $\ell_2$ coefficient, often leads to better outlier detection performance, but that comes at the cost of repair quality due to the VAE collapsing to mean behaviour. For each model, the hyperparameter search was carried out for each dataset at a noise level of 35%, and with 25 samples per class.

- **VAE-L2** In *Synthetic-Shapes*: 200 epochs; KL divergence annealing used; $\ell_2$ coefficient is 35.0. In *Frey-Faces*: 300 epochs; KL divergence annealing used; $\ell_2$ coefficient is 100.0. In *Fashion-MNIST*: 100 epochs; KL divergence annealing used; $\ell_2$ coefficient is 100.0.

- **VAEGMM** In *Synthetic-Shapes*: 200 epochs; KL divergence annealing used; fraction of clean data $\alpha = 0.6$; (trusted set) up-sampling coefficient $\beta = 1000$; $\sigma_{y=1} = 0.9$ and $\sigma_{y=0} = 5.0$. In *Frey-Faces*: 300 epochs; KL divergence annealing used; fraction of clean data $\alpha = 0.6$; (trusted set) up-sampling coefficient $\beta = 1000$; $\sigma_{y=1} = 0.6$ and $\sigma_{y=0} = 5.0$. In *Fashion-MNIST*: 100 epochs; KL divergence annealing used; fraction of clean data $\alpha = 0.6$; (trusted set) up-sampling coefficient $\beta = 100$; $\sigma_{y=1} = 0.5$ and $\sigma_{y=0} = 5.0$.

- **CVAE** In *Synthetic-Shapes*: 200 epochs; KL divergence annealing used; $\sigma = 0.5$. In *Frey-Faces*: 300 epochs; KL divergence annealing used; $\sigma = 0.2$. In *Fashion-MNIST*: 100 epochs; KL divergence annealing used; $\sigma = 0.5$.

- **CCVAE** In *Synthetic-Shapes*: 200 epochs; fraction of clean data $\alpha = 0.6$; (trusted set) up-sampling coefficient $\beta = 50000.0$. In *Frey-Faces*: 300 epochs; fraction of clean data $\alpha = 0.6$; (trusted set) up-sampling coefficient $\beta = 10000.0$. In *Fashion-MNIST*: 100 epochs; fraction of clean data $\alpha = 0.6$; (trusted set) up-sampling coefficient $\beta = 250000.0$.

- **CLSVAE** In *Synthetic-Shapes*: 200 epochs; fraction of clean data $\alpha = 0.6$; KL divergence annealing used; (trusted set) up-sampling coefficient $\beta = 1000.0$. $\sigma_\epsilon = 0.5$; $\sigma_c = 0.5$; $\sigma_d = 5.0$; distance correlation (DC) penalty used; $\lambda_t$ annealing ratio of 0.5 (DC penalty); $\lambda_T$ maximum value of 100.0 (DC penalty). In *Frey-Faces*: 300 epochs; fraction of clean data $\alpha = 0.6$; KL divergence annealing used; (trusted set) up-sampling coefficient $\beta = 1000.0$. $\sigma_\epsilon = 0.6$; $\sigma_c = 0.2$; $\sigma_d = 5.0$; distance correlation (DC) penalty used; $\lambda_t$ annealing ratio of 0.5 (DC penalty); $\lambda_T$ maximum value of 1000.0 (DC penalty). In *Fashion-MNIST*: 100 epochs; fraction of clean data $\alpha = 0.6$; KL divergence annealing used; (trusted set) up-sampling coefficient $\beta = 100.0$. $\sigma_\epsilon = 0.1$; $\sigma_c = 0.2$; $\sigma_d = 5.0$; distance correlation (DC) penalty used; $\lambda_t$ annealing ratio of 0.5 (DC penalty); $\lambda_T$ maximum value of 1000.0 (DC penalty).

- **CLSVAE-NODC** Same hyperparameter options as **CLSVAE** but without the distance correlation penalty.

Note the annealing ratio above for the KL divergence and for the DC penalty is inspired by Fu et al. (2019). We use only one cycle (monotonic), and the $R$ (or *ratio*) is the proportion used to increase the penalty coefficient (or KL term coefficient) – e.g. 0.5.

**Optimization**

We used the PyTorch framework to code all our models, and trained on a GeForce TITAN X GPU. All models were trained using the Adam optimizer, with an initial learning rate of 0.001.

**Model Architectures**

For continuous type data, i.e. *Fashion-MNIST* and *Frey-Faces*, we used the Gaussian distribution as the likelihood of each pixel in the reconstruction loss. The variance of the Gaussian distribution is shared amongst all the pixels in the image, and it is learnt as a parameter of the model. This was done for all models. For binary type data, i.e. *Synthetic-Shapes*, we treated each pixel as a Bernoulli variable, and used the log-likelihood of this distribution for each pixel in the reconstruction loss. This was done for all models.

We used very similar encoder and decoder architectures for all models, so as to be fair and the results comparable. In the case of CLSVAE we used two encoders, one for the clean subspace $z_c$, and the other for the dirty subspace $z_d$. This architecture yielded better results for us in terms of repair in the trusted set. CLSVAE architecture can be seen in Table 5 for the encoders and decoder, and the classifier can be seen in 7. In Table 2, we see the neural architecture for the encoder and decoder of VAE and VAEGMM. The classifier architecture for the VAEGMM can be seen in Table 6. In Table

3, we find the architecture for the encoder and decoder of CVAE. In Table 4, we find the architecture for the encoder and decoder of CCVAE. The classifier architecture of CCVAE is just $z_c$ multiplied by a weight parameter plus a bias parameter, and then a *sigmoid* non-linearity is applied – like in Joy et al. (2020).

| Encoder | Decoder |
|---|---|
| (img_size, 200) $\to$ | $\to$ (15, 50) $\to$ |
| $\to$ ReLU $\to$ | $\to$ ReLU $\to$ |
| $\to$ (200, 100) $\to$ | $\to$ (50, 100) $\to$ |
| $\to$ ReLU $\to$ | $\to$ ReLU $\to$ |
| $\to$ (100, 50) $\to$ | $\to$ (100, 200) $\to$ |
| $\to$ ReLU $\to$ | $\to$ ReLU $\to$ |
| $\to 2 \times$ (50, 15) | $\to$ (200, img_size) |

Table 2: Architecture of encoder and decoder for VAE and VAEGMM. Further, for binary pixels the decoder will use a Sigmoid non-linearity at the end.

| Encoder | Decoder |
|---|---|
| (img_size, 200) $\to$ | $\to$ (16, 50) $\to$ |
| $\to$ ReLU $\to$ | $\to$ ReLU $\to$ |
| $\to$ (200, 100) $\to$ | $\to$ (50, 100) $\to$ |
| $\to$ ReLU $\to$ | $\to$ ReLU $\to$ |
| $\to$ (100, 50) $\to$ | $\to$ (100, 200) $\to$ |
| $\to$ ReLU $\to$ | $\to$ ReLU $\to$ |
| $\to 2 \times$ (50, 15) | $\to$ (200, img_size) |

Table 3: Architecture of encoder and decoder for CVAE. Further, for binary pixels the decoder will use a Sigmoid non-linearity at the end.

| Encoder | Decoder |
|---|---|
| (img_size, 200) $\to$ | $\to$ (16, 50) $\to$ |
| $\to$ ReLU $\to$ | $\to$ ReLU $\to$ |
| $\to$ (200, 100) $\to$ | $\to$ (50, 100) $\to$ |
| $\to$ ReLU $\to$ | $\to$ ReLU $\to$ |
| $\to$ (100, 50) $\to$ | $\to$ (100, 200) $\to$ |
| $\to$ ReLU $\to$ | $\to$ ReLU $\to$ |
| $\to 2 \times$ (50, 16) | $\to$ (200, img_size) |

Table 4: Architecture of encoder and decoder for CCVAE. Further, for binary pixels the decoder will use a Sigmoid non-linearity at the end.

| Encoder $z_c$ | Encoder $z_d$ | Decoder |
|---|---|---|
| (img_size, 200) $\to$ | (img_size, 200) $\to$ | $\to$ (15, 50) $\to$ |
| $\to$ ReLU $\to$ | $\to$ ReLU $\to$ | $\to$ ReLU $\to$ |
| $\to$ (200, 100) $\to$ | $\to$ (200, 100) $\to$ | $\to$ (50, 100) $\to$ |
| $\to$ ReLU $\to$ | $\to$ ReLU $\to$ | $\to$ ReLU $\to$ |
| $\to$ (100, 50) $\to$ | $\to$ (100, 50) $\to$ | $\to$ (100, 200) $\to$ |
| $\to$ ReLU $\to$ | $\to$ ReLU $\to$ | $\to$ ReLU $\to$ |
| $\to 2 \times$ (50, 10) | $\to 2 \times$ (50, 5) | $\to$ (200, img_size) |

Table 5: Architecture of encoder and decoder for CLSVAE. Note for CLSVAE latent space of size 15 is split: 10 for $z_c$ and 5 for $z_d$. Further, for binary pixels the decoder will use a Sigmoid non-linearity at the end.

| Classifier |
|---|
| (img_size, 200) $\rightarrow$ |
| ReLU |
| $\rightarrow$ (200, 100) $\rightarrow$ |
| ReLU |
| $\rightarrow$ (100, 50) $\rightarrow$ |
| ReLU |
| $\rightarrow$ (50, 1) |
| Sigmoid |

Table 6: Architecture of classifier VAEGMM.

| Classifier |
|---|
| (15, 7) $\rightarrow$ |
| ReLU |
| $\rightarrow$ (7, 5) $\rightarrow$ |
| ReLU |
| $\rightarrow$ (5, 1) |
| Sigmoid |

Table 7: Architecture of classifier CLSVAE, input is $z$.

# I    LARGER VERSION OF RECONSTRUCTIONS (REPAIRS) FIGURE

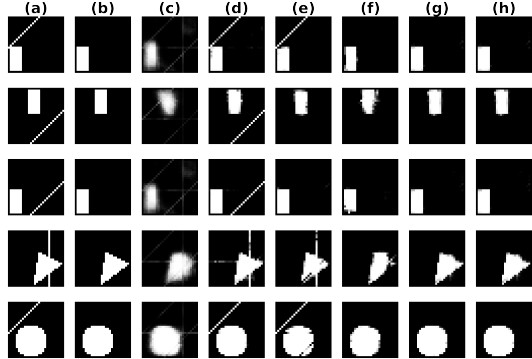

(a) *Synthetic-Shapes*: 35% noise, 10 labels per class
(1.6% of dataset).

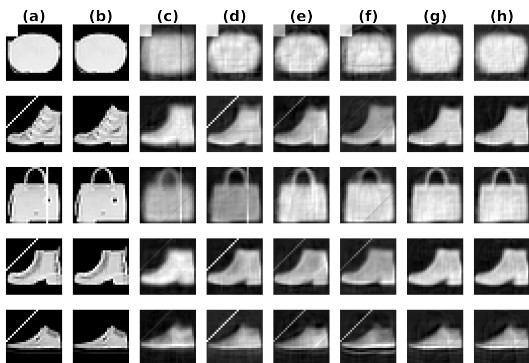

(b) *Fashion-MNIST*: 35% noise, 10 labels per class
(0.25% of dataset).

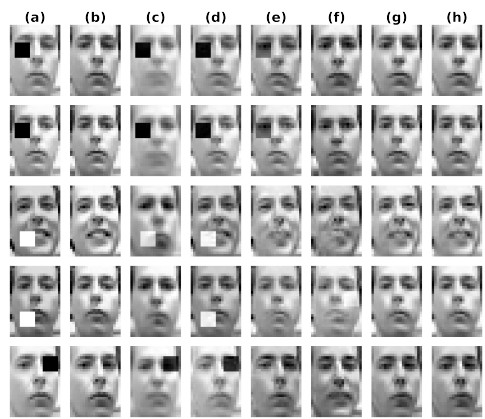

(c) *Frey-Faces*: 35% noise, 10 labels per class
(2.5% of dataset)

Figure 3: Images for model repair (reconstruction), outlier (corrupted) and inlier (uncorrupted): (a) Original (Outlier); (b) Ground-Truth (Inlier); (c) VAE-L2; (d) VAEGMM; (e) CVAE; (f) CCVAE; (g) CLSVAE-NODC; (h) CLSVAE.

# J    RESULTS FOR ALL NOISE LEVELS AND TRUSTED SET SIZES (ALL SWEEPS)

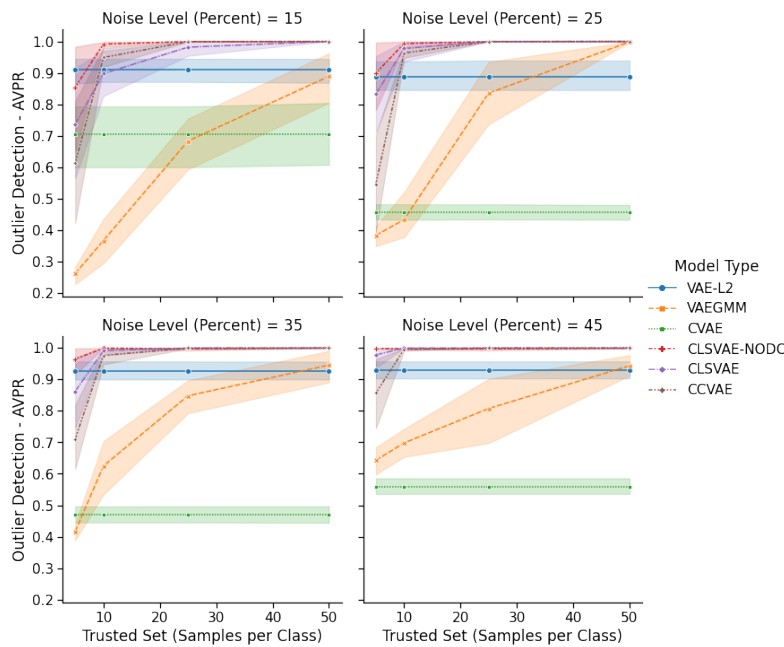

Figure 4: *Synthetic-Shapes*. Outlier detection (AVPR) where *higher is better*. Trusted set range sweep where $\text{TS}_{\text{size}} = [5, 10, 25, 50]$ samples per class, i.e. $[0.8\%, 1.6\%, 4\%, 8\%]$ of the entire dataset.

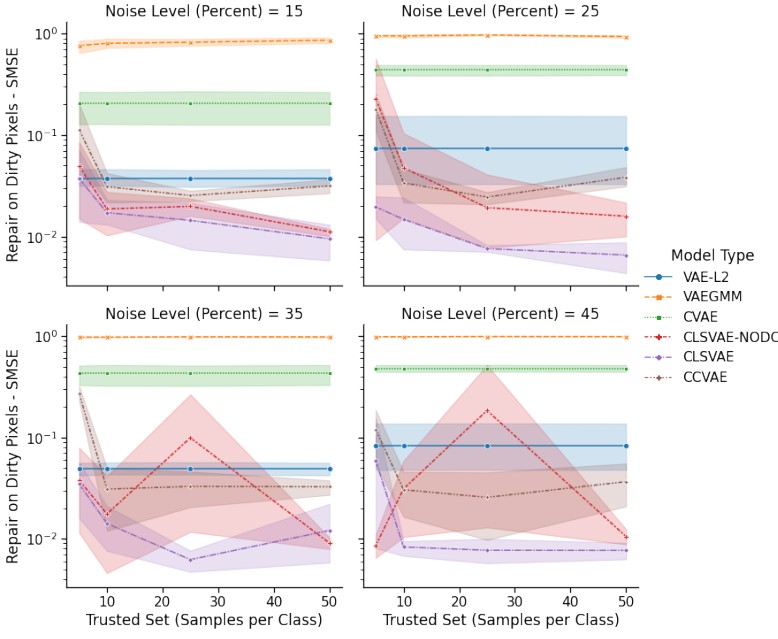

Figure 5: *Synthetic-Shapes*. Repair of dirty pixels in outliers (SMSE), where *lower is better*. Trusted set range sweep where $\text{TS}_{\text{size}} = [5, 10, 25, 50]$ samples per class, i.e. $[0.8\%, 1.6\%, 4\%, 8\%]$ of the entire dataset.

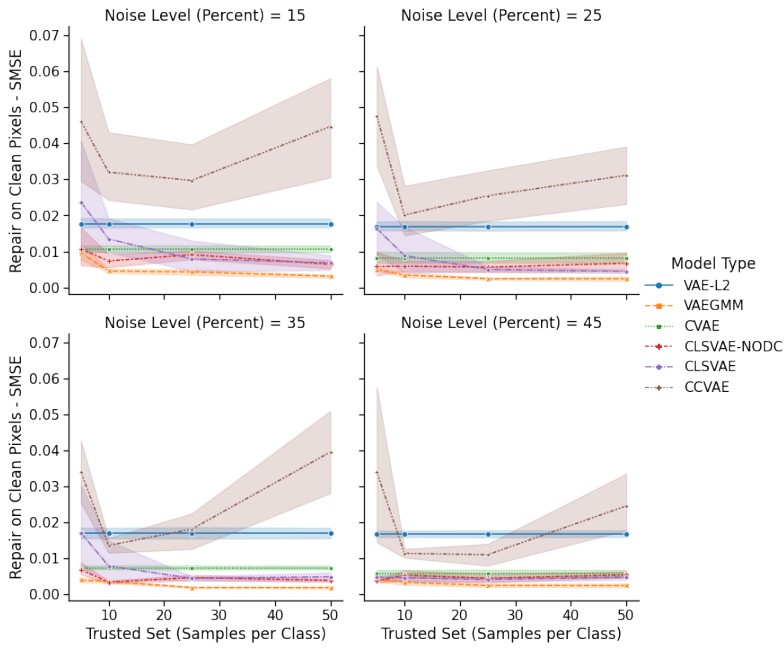

Figure 6: *Synthetic-Shapes*. Repair of clean pixels in outliers (SMSE), i.e. distortion, where *lower is better*. Trusted set range sweep where $TS_{size} = [5, 10, 25, 50]$ samples per class, i.e. $[0.8\%, 1.6\%, 4\%, 8\%]$ of the entire dataset.

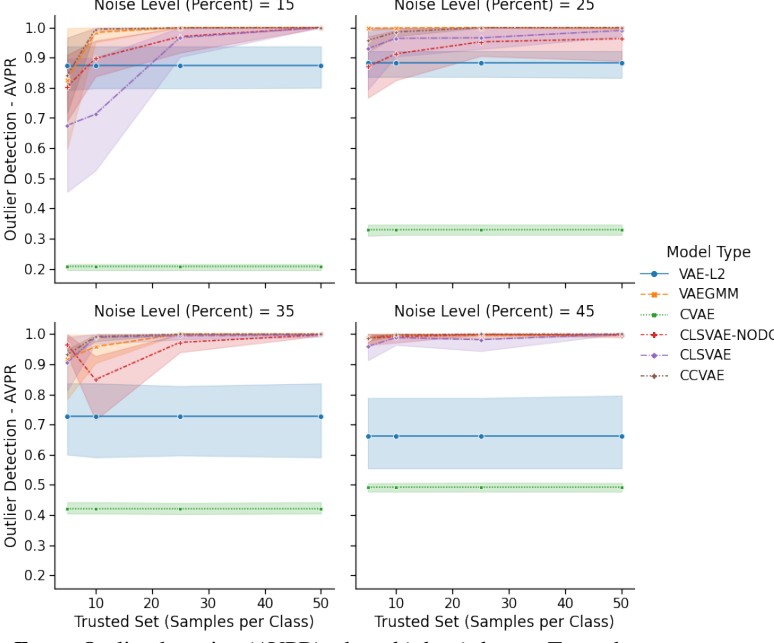

Figure 7: *Frey-Faces*. Outlier detection (AVPR) where *higher is better*. Trusted set range sweep where $TS_{size} = [5, 10, 25, 50]$ samples per class, i.e. $[1.3\%, 2.5\%, 6.4\%, 12.7\%]$ of the entire dataset.

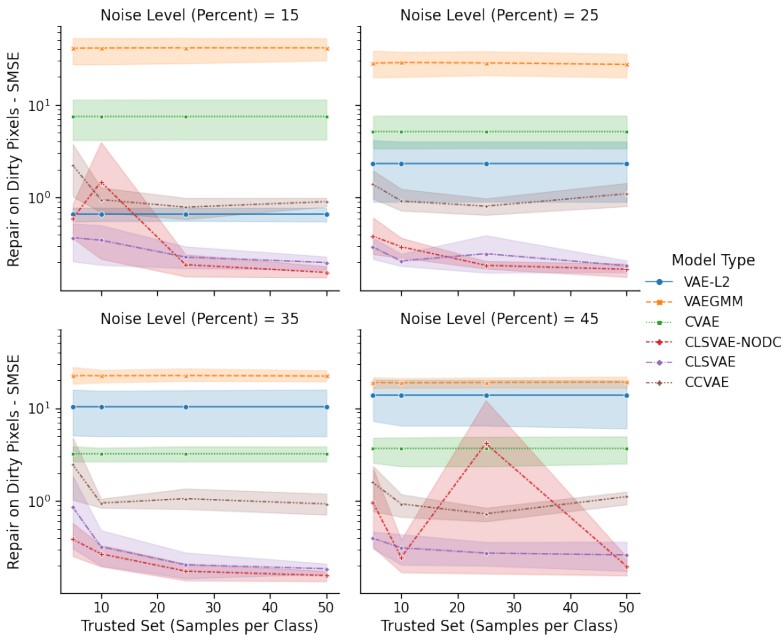

Figure 8: *Frey-Faces*. Repair of dirty pixels in outliers (SMSE), where *lower is better*. Trusted set range sweep where $TS_{size} = [5, 10, 25, 50]$ samples per class, i.e. $[1.3\%, 2.5\%, 6.4\%, 12.7\%]$ of the entire dataset.

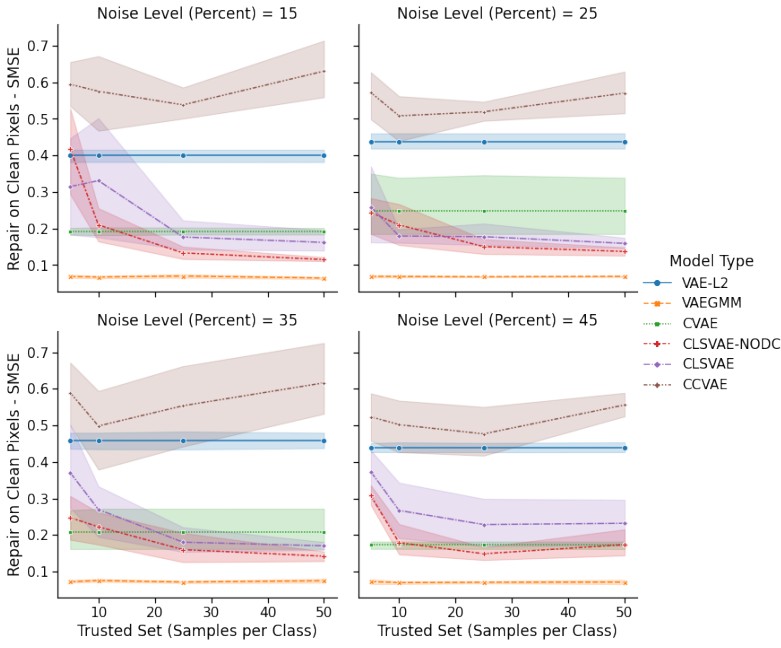

Figure 9: *Frey-Faces*. Repair of clean pixels in outliers (SMSE), i.e. distortion, where *lower is better*. Trusted set range sweep where $TS_{size} = [5, 10, 25, 50]$ samples per class, i.e. $[1.3\%, 2.5\%, 6.4\%, 12.7\%]$ of the entire dataset.

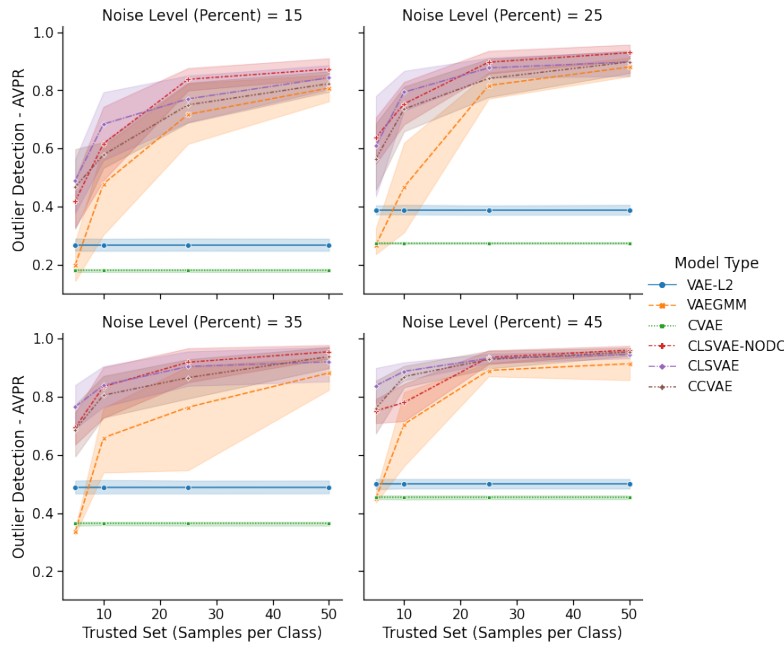

Figure 10: *Fashion-MNIST*. Outlier detection (AVPR) where *higher is better*. Trusted set range sweep where $\text{TS}_{\text{size}} = [5, 10, 25, 50]$ samples per class, i.e. $[0.12\%, 0.25\%, 0.64\%, 1.28\%]$ of the entire dataset.

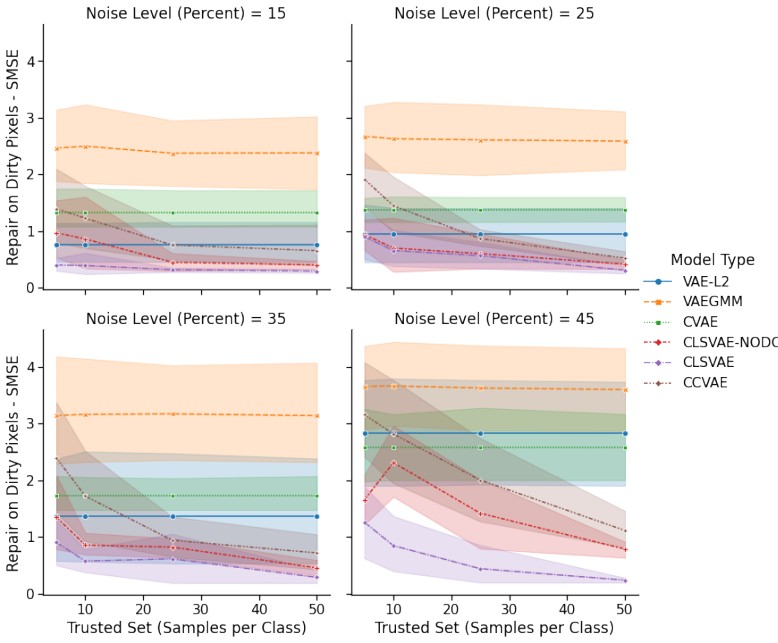

Figure 11: *Fashion-MNIST*. Repair of dirty pixels in outliers (SMSE), where *lower is better*. Trusted set range sweep where $\text{TS}_{\text{size}} = [5, 10, 25, 50]$ samples per class, i.e. $[0.12\%, 0.25\%, 0.64\%, 1.28\%]$ of the entire dataset.

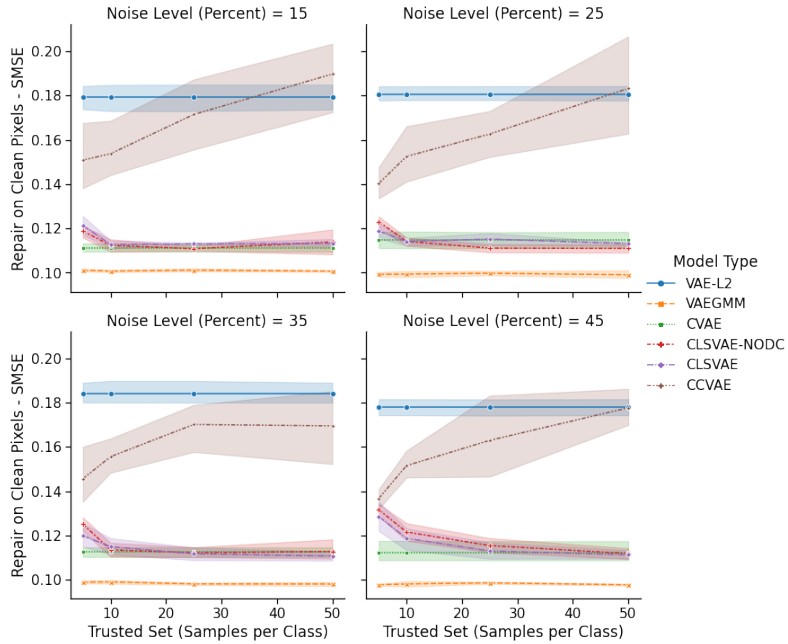

Figure 12: *Fashion-MNIST*. Repair of clean pixels in outliers (SMSE), i.e. distortion, where *lower is better*. Trusted set range sweep where $TS_{size} = [5, 10, 25, 50]$ samples per class, i.e. $[0.12\%, 0.25\%, 0.64\%, 1.28\%]$ of the entire dataset.

# K  ADDITIONAL RECONSTRUCTIONS (REPAIRS) FOR ALL DATASETS

## K.1  SYNTHETIC-SHAPES

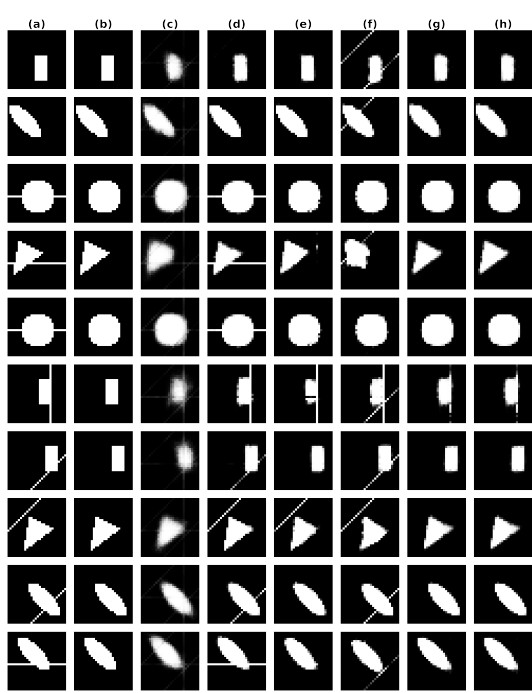

Figure 13: Images for model repair (reconstruction), outlier (corrupted) and inlier (uncorrupted): (a) Original (Outlier); (b) Ground-Truth (Inlier); (c) VAE-L2; (d) VAEGMM; (e) CVAE; (f) CCVAE; (g) CLSVAE-NODC; (h) CLSVAE. **The first two rows are inlier examples**, the others being outliers. *Synthetic-Shapes*: **35% noise, 5 labels per class** (0.8% of dataset).

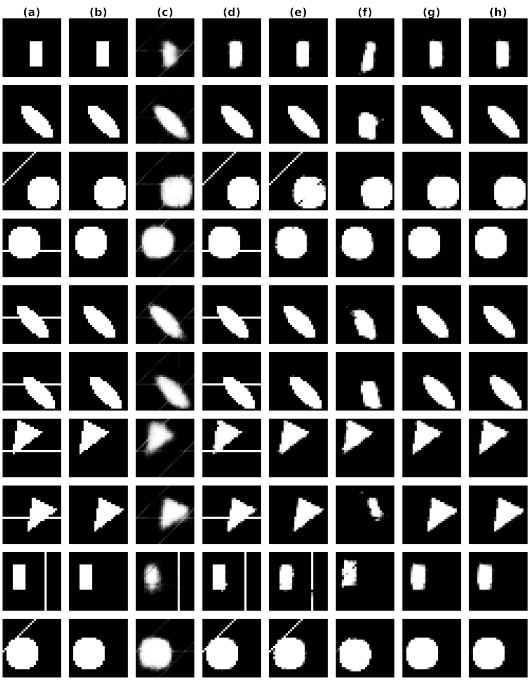

Figure 14: Images for model repair (reconstruction), outlier (corrupted) and inlier (uncorrupted): (a) Original (Outlier); (b) Ground-Truth (Inlier); (c) VAE-L2; (d) VAEGMM; (e) CVAE; (f) CCVAE; (g) CLSVAE-NODC; (h) CLSVAE. **The first two rows are inlier examples**, the others being outliers. *Synthetic-Shapes*: **35% noise**, **50 labels per class** (8% of dataset).

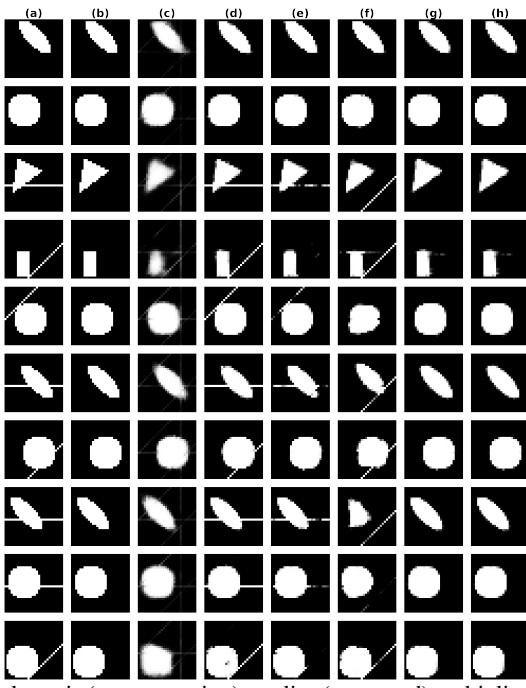

Figure 15: Images for model repair (reconstruction), outlier (corrupted) and inlier (uncorrupted): (a) Original (Outlier); (b) Ground-Truth (Inlier); (c) VAE-L2; (d) VAEGMM; (e) CVAE; (f) CCVAE; (g) CLSVAE-NODC; (h) CLSVAE. **The first two rows are inlier examples**, the others being outliers. *Synthetic-Shapes*: **45% noise**, **5 labels per class** (0.8% of dataset).

## K.2   FREY-FACES

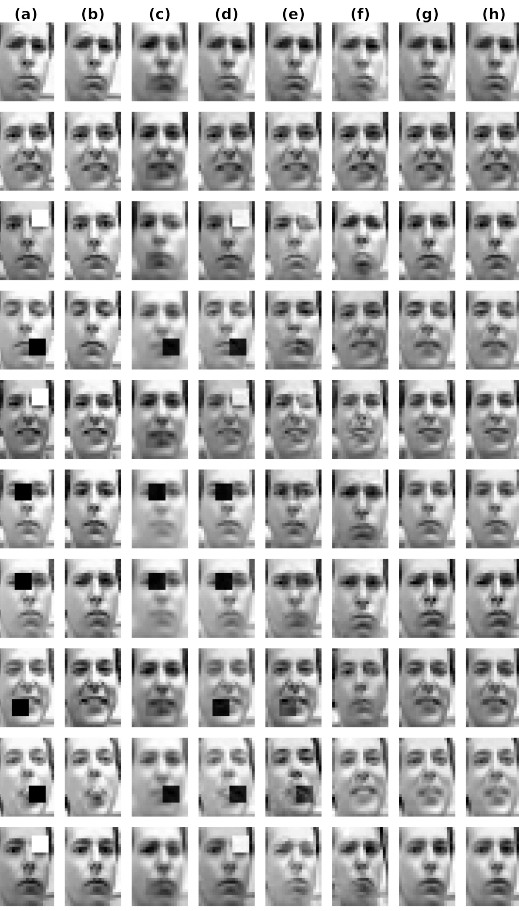

Figure 16: Images for model repair (reconstruction), outlier (corrupted) and inlier (uncorrupted): (a) Original (Outlier); (b) Ground-Truth (Inlier); (c) VAE-L2; (d) VAEGMM; (e) CVAE; (f) CCVAE; (g) CLSVAE-NODC; (h) CLSVAE. **The first two rows are inlier examples**, the others being outliers. *Frey-Faces*: **35% noise**, **10 labels per class** (2.5% of dataset).

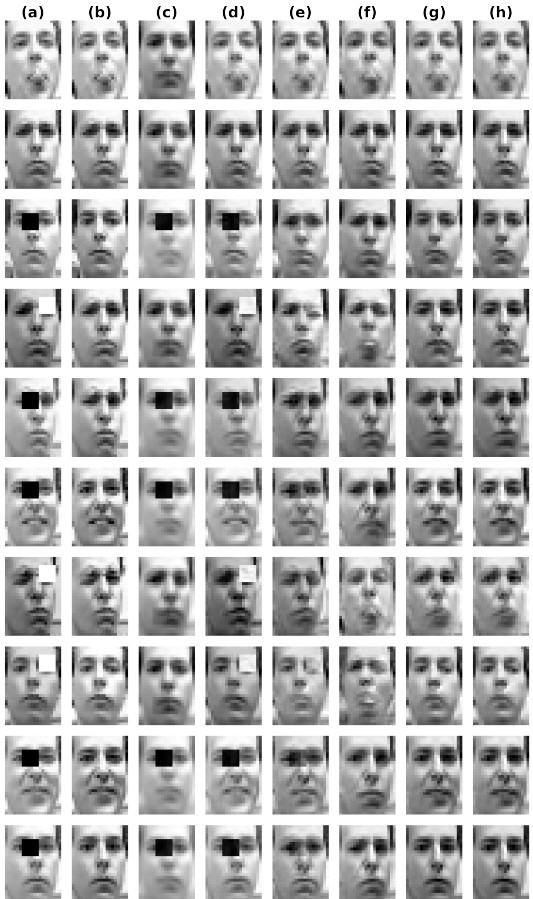

Figure 17: Images for model repair (reconstruction), outlier (corrupted) and inlier (uncorrupted): (a) Original (Outlier); (b) Ground-Truth (Inlier); (c) VAE-L2; (d) VAEGMM; (e) CVAE; (f) CCVAE; (g) CLSVAE-NODC; (h) CLSVAE. **The first two rows are inlier examples**, the others being outliers. *Frey-Faces*: **35% noise**, **50 labels per class** (12.7% of dataset).

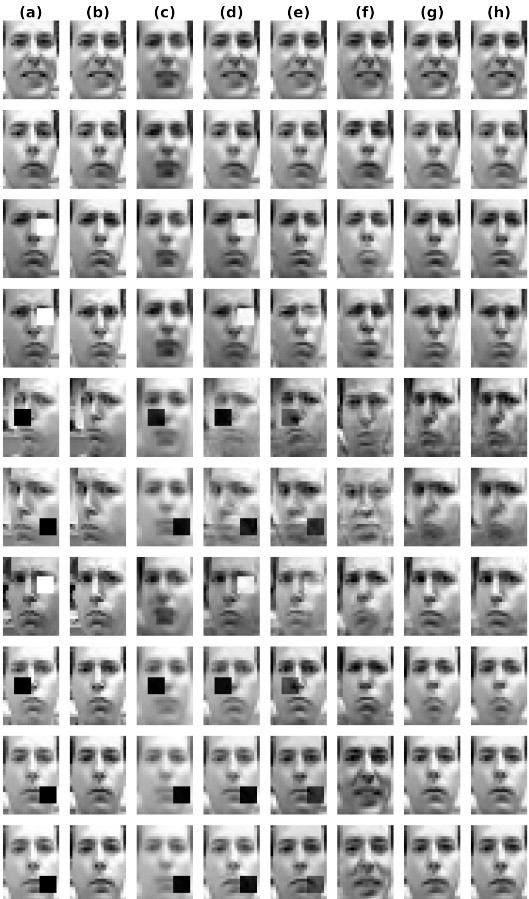

Figure 18: Images for model repair (reconstruction), outlier (corrupted) and inlier (uncorrupted): (a) Original (Outlier); (b) Ground-Truth (Inlier); (c) VAE-L2; (d) VAEGMM; (e) CVAE; (f) CCVAE; (g) CLSVAE-NODC; (h) CLSVAE. **The first two rows are inlier examples**, the others being outliers. *Frey-Faces*: **45% noise**, **10 labels per class** (2.5% of dataset).

## K.3 FASHION-MNIST

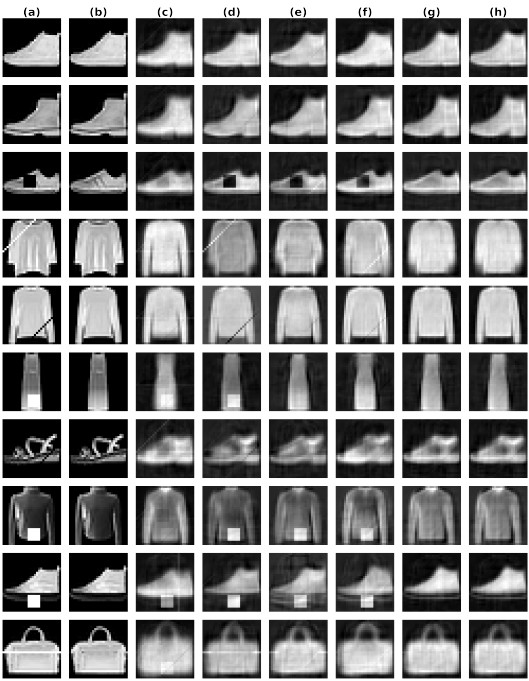

Figure 19: Images for model repair (reconstruction), outlier (corrupted) and inlier (uncorrupted): (a) Original (Outlier); (b) Ground-Truth (Inlier); (c) VAE-L2; (d) VAEGMM; (e) CVAE; (f) CCVAE; (g) CLSVAE-NODC; (h) CLSVAE. **The first two rows are inlier examples**, the others being outliers. *Fashion-MNIST*: **35% noise, 10 labels per class** (0.25% of dataset).

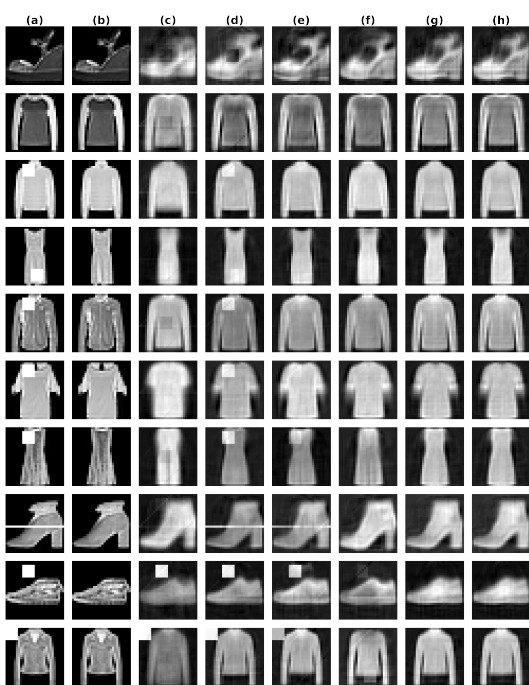

Figure 20: Images for model repair (reconstruction), outlier (corrupted) and inlier (uncorrupted): (a) Original (Outlier); (b) Ground-Truth (Inlier); (c) VAE-L2; (d) VAEGMM; (e) CVAE; (f) CCVAE; (g) CLSVAE-NODC; (h) CLSVAE. **The first two rows are inlier examples**, the others being outliers. *Fashion-MNIST*: **35% noise, 50 labels per class** (1.28% of dataset).

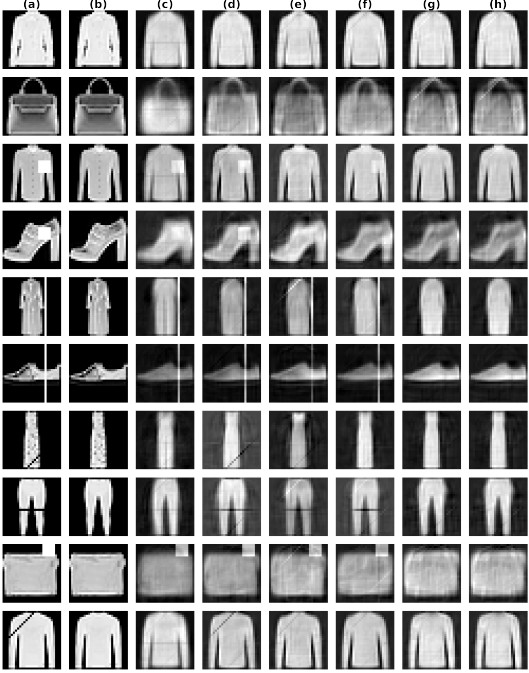

Figure 21: Images for model repair (reconstruction), outlier (corrupted) and inlier (uncorrupted): (a) Original (Outlier); (b) Ground-Truth (Inlier); (c) VAE-L2; (d) VAEGMM; (e) CVAE; (f) CCVAE; (g) CLSVAE-NODC; (h) CLSVAE. **The first two rows are inlier examples**, the others being outliers. *Fashion-MNIST*: **45% noise**, **10 labels per class** (0.25% of dataset).

## L    TESTING COMPRESSION HYPOTHESIS: ENTROPY OF CLEAN VS. CORRUPTED DATA

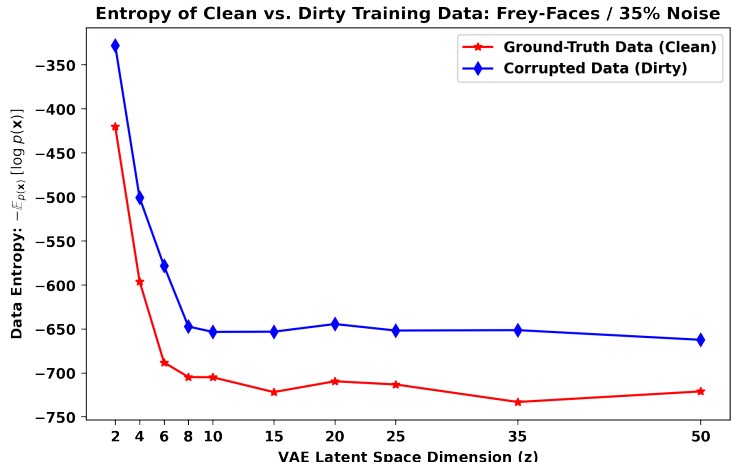

Figure 22: Entropy of ground-truth training data (clean: without corruption) vs the entropy of corrupted training data (as in Table 1). Entropy estimation via IWAE (Burda et al., 2016), using a standard VAE (not regularized). VAE uses same architecture of Annex H, except dimension of **z** (latent space) now has the range [2, 4, 6, 8, 10, 15, 20, 25, 35, 50] (x-axis). *Frey-Faces* training data, with 35% noise level for corrupted dataset.

In this section, we experimentally compare the entropy of clean (without corruption) training data, and the entropy of corrupted training data. A larger entropy means that a dataset has larger variance overall. Note that in the setup of our problem, i.e. repairing systematic errors (see Section 3), only corrupted data is used for training. In Figure 22, for *Frey-Faces*, we compare the estimated entropy of clean training data against one that has been corrupted (35 % noise, corruption as in Table 1).

We estimate the entropy by first training a standard VAE model on the dataset (clean or corrupted), and then after training, we compute a tight bound on the marginal log-likelihood of that dataset. We compute this tight bound via IWAE estimator (importance weighted autoencoders, (Burda et al., 2016)), where we use $K = 250$ samples. Note that entropy is $\mathcal{H}(x) = -\mathbb{E}_{p_\theta(x)}\left[\log p_\theta(x)\right]$, and hence marginal log-likelihood is just $-\mathcal{H}(x)$. We vary the dimension of VAE latent space ($\mathbf{z}$) in order to see how well the model can learn the training data. For this experiment, the VAE is not regularized.

The main idea is to prove empirically that corrupted data has larger variance than clean data, and hence has more information to be modelled. Specifically, when using the type of masking corruptions used in our experiments. If a dataset has larger estimated entropy, then the data can be seen has having larger variance. A dataset with larger variance is a dataset with more diversity in terms of the patterns it contains, and hence it has more information to be compressed (or modelled). In particular, in a VAE this means more capacity (e.g. larger latent space) is needed to model a larger variance dataset.

Our claim, supported in literature (Eduardo et al., 2020; Ruff et al., 2019), is that a dataset that has been corrupted has larger variance because the added outliers (e.g. systematic errors) increase data pattern diversity, hence increasing entropy. Looking at Figure 22, we see that overall the entropy of corrupted data is larger than clean data, for all sizes of $\mathbf{z}$. Hence, corrupted data has larger variance than clean data. For the smaller dimensions of $\mathbf{z}$, in range $[2, 10]$ units, we see that the VAE has less trouble learning the clean data compared to the corrupted data. This is also evidenced by how more quickly the entropy decreases for clean data relative to corrupt data, as we increase the latent space size in $[2, 10]$. We conclude that a VAE only needs a smaller latent space (subspace) to model clean data (inliers), and that corrupted data needs a larger latent space to be modelled properly.

