# OpenReview forum: "Repairing Systematic Outliers by Learning Clean Subspaces in VAEs"
_ICLR.cc/2022/Conference — ICLR 2022 Submitted_

### Official Review · Reviewer_5JVH · 2021-10-29

**Correctness:** 4
**Technical Novelty And Significance:** 2
**Empirical Novelty And Significance:** 2
**Recommendation:** 6
**Confidence:** 2

**Main Review:**

Strength:
- The paper is well motivated and generally well written. The key idea of subspace learning is intuitive and easy to understand.
- Results look to be good and support the claim.

Weaknesses:
- While I found Section. 3 to be well written, but Section 4, which contains the math and equations, seems to be difficult to follow, especially for a person that is not necessarily familiar with VAEs and generative models. Perhaps a simple introduction of those could help.

- An issue of the paper, from my perspective, is that the experiments are simulated and not real. Therefore, I am not fully convinced that the developed method will have practical impact.  In particular, is it possible to test the method on some dataset that is well-known to be corrupted by systematic outliers (rather than simulating the outliers)?  One example that I am aware of, is that, in robot perception, wrong correspondences is a big challenge. For example, in point cloud registration, establishing reliable matches between a pair of point clouds is a very challenging task. There is a huge amount of literature on this problem, but perhaps [ref1] and [ref2] could be good references.  This maybe a future work direction for the authors to consider.

- I also think an ablation study could help, where the impact of the size of the trusted set on the performance could be investigated. If the authors increase the size of the trusted set from <10% to 50%, do we expect a gain in the performance?


[Ref1] Yang, Heng, Jingnan Shi, and Luca Carlone. "Teaser: Fast and certifiable point cloud registration." IEEE Transactions on Robotics 37, no. 2 (2020): 314-333.

[Ref2] Yi, Kwang Moo, Eduard Trulls, Yuki Ono, Vincent Lepetit, Mathieu Salzmann, and Pascal Fua. "Learning to find good correspondences." In Proceedings of the IEEE conference on computer vision and pattern recognition, pp. 2666-2674. 2018.

**Summary Of The Paper:**

This paper develops a deep learning based method for detecting and repairing systematic outliers in data (before performing model learning on the data).

The method is called Clean Subspace Variational Autoencoder (CLSVAE). CLSVAE takes as input a trusted set provided by the user (the trusted set contains a set of inliers and outliers, but not requiring information about how the outliers are corrupted), and seeks to learn a variational autoencoder where the latent code is a concatenation of two subspaces, a clean subspace and a dirty subspace. CLSVAE is semi-supervised because the labelled trusted set is much smaller than the unlabelled set.

The paper performs outlier detection and repair benchmarks on three image datasets with systematic outliers, and shows that CLSVAE achieves favorable results.



**Summary Of The Review:**

I am not an expert in this field. I tend to weak accept this paper, but I will also see if other reviewers have serious concerns.

---

> ### Author Response · Authors · 2021-11-21
> **Answer to Review**
>
> We want to thank the reviewer for the time spent on our paper, thoughtful comments and questions, and the constructive feedback. We answer some of your questions and concerns below.
>
> Rev3 (code: 5JVH)
>
> **Q1:
> Section 4 contains math and equations, which seems difficult to follow, especially for a person not familiar with VAEs and generative models.**
>
> **R1:**
> We will try and add a reference to work (papers) explaining VAE models, or survey. In addition, a standard VAE (using L2 regularization) is presented in Annex A. **We added a sentence in Section 4 about this, also pointing to Annex A (highlighted in blue).**
>
> **Q2:
> (a) An issue of the paper, from my perspective, is that the experiments are simulated and not real. Therefore, I am not fully convinced that the developed method will have practical impact. In particular, is it possible to test the method on some dataset that is well-known to be corrupted by systematic outliers (rather than simulating the outliers)?**
>
> **(b) For example, in point cloud registration, establishing reliable matches between a pair of point clouds is a very challenging task. There is a huge amount of literature on this problem, but perhaps [ref1] and [ref2] could be good references. This may be a future work direction for the authors to consider.**
>
> **R2:**
>
> **(a) About practical impact and synthetic error injection:**
>
> Please see our answer to this in a top-level comment (named **TPL-1**), for the benefit of all reviewers.
> We will further address this in the camera ready paper (if accepted).
>
> **(b) About the suggested task (i.e. establishing reliable matches between a pair of point clouds)**
>
> Thank you for this suggestion! We agree that it could be interesting to explore as an extension to our current model. We also agree that this problem can be seen as one type of systematic corruption, or transformation, which is related to translation or offset errors [Taylor, 1997].
>
> **Q3:
> I also think an ablation study could help, where the impact of the size of the trusted set on the performance could be investigated. If the authors increase the size of the trusted set from <10% to 50%, do we expect a gain in the performance?**
>
> **R3:**
> We would like to point out that we carried out such a study in Annex J, studying the impact of the size of the trusted set. For Fashion-MNIST we also presented part of those results in the main paper, see Figure 2 (a). We used a range that we thought was realistic enough for a user to label efficiently (in Section 5). Moreover, in Annex J we can see the impact of the trusted set size for different models, datasets, and corruption levels.
> There is another reason why we did not try 50% of labelled data. In Annex J, if we look at our model (CLSVAE) we see that performance (detection and repair) is close to stabilizing, or stabilized already, where the trusted set is largest (seen also in Figure 2(a)). This is clearer in outlier detection. Overall, we estimate the gains to be modest or marginal in terms of performance at 50%. In addition, we see that our best performing baseline model (CCVAE) is also close to stabilizing, or stabilized, where the trusted set is largest.
> In Annex J, we see that both CLSVAE and CCVAE increase their performance gains in both outlier detection (higher AVPR) and data repair (lower SMSE) as the trusted set increases. Note CVAE is fully supervised so its performance is static.

---

### Official Review · Reviewer_7fHt · 2021-11-02

**Correctness:** 2
**Technical Novelty And Significance:** 2
**Empirical Novelty And Significance:** 2
**Recommendation:** 6
**Confidence:** 4

**Main Review:**

Overall, the paper is well written and the contributions are clearly formulated.
However, I have several concerns with it:
- The intuition behind the assumption that inliers can be represented in a subspace of the clean and dirty patterns is not so obvious, especially given the datasets that have been used in the experiments. These systematic errors of adding or masking out lines or squares at fixed positions actually "simplify" the data, and the variance is reduced.
- The problem is clearly stated, but to me it is not evident what would be a practical application of this algorithm and a real use case. First of all, the fact that a labelled (trusted) data set is needed (albeit small) constrains the practical usage. Second, the use case of detecting images where some pixels are set to default values (e.g. from a deficient camera) would be dependent on a specific sensor. Finally, the application of watermarks is mentioned and makes more sense. However, the experimental evaluation does not include this more complex type of outliers.
- There is an extensive appendix, which would be Ok. But the main paper misses some explanations and details that are only mentioned in the apendices (for example appendix E and F)

Some minor remarks:
- Section 4.2. on the variational model mixes the theoretical model with some implementation aspects, like "stop gradient" (which is only related to training/optimisation). Also some parts are not explained, like the distribution "pi".
- The figures and tables are too small.


**Summary Of The Paper:**

This paper presents a neural network model based on Variational Autoencoders (VAE) that learn an implicit representation separating outliers with systematic "errors" from inliers using a small labelled subset of the training data set (trusted set).
More specifically, clean data and recurring systematic errors are represented in separate latent subspaces, where outliers are supposed to be a (linear) combination of such clean and "dirty" patterns.
After training the model, there are two tasks: outlier detection and automated repair, removing the "dirty" pattern(s) from the detected outliers.


**Summary Of The Review:**

The paper presents and interested approach that is well formulated. However, the main motivations and assumptions are not well comprehensible, theoretical and practical implementation aspects are mixed, and the presentation is lacking clarity in some places.

---

> ### Author Response · Authors · 2021-11-21
> **Answer to Review (Part 1)**
>
> We want to thank the reviewer for the time spent on our paper, thoughtful comments and questions, and the constructive feedback. We answer some of your questions and concerns below.
>
> **NOTE: additional references are in Answer Part-3 (i.e. for citations).**
>
> **Q1:
> The intuition behind the assumption that inliers can be represented in a subspace of the clean and dirty patterns is not so obvious, especially given the datasets that have been used in the experiments. These systematic errors of adding or masking out lines or squares at fixed positions actually "simplify" the data, and the variance is reduced.**
>
> **R1:**
> We would like to point out that we conducted some *new experiments* to show empirically that: (a) corrupted data has larger entropy than clean data, and hence larger variance; (b) clean data can be learnt better by smaller latent space VAEs than corrupted data. We postulate that (b) follows from (a): since clean data has lower variance (entropy), which means it has less pattern diversity (no systematic errors), hence a smaller code can be used for data encoding. We conducted these experiments using the same type of noise found in Table 1 of our manuscript (for Frey-Faces dataset), and we used a standard VAE model. **These new results and discussion can be found in Annex L.**
>
> Moreover, the modelling assumption that corrupted data usually has larger variance than its clean counterpart is standard in outlier detection literature. Models based on this assumption have been shown to be effective, for instance: [Eduardo et al 2020] using a large variance Gaussian for outliers; [Ruff et al., 2021] defining a larger entropy (or variance) for outliers in latent space;  [r2_ref_1] where random errors are injected using masks of objects (i.e. circles) for grey-level images, and this model assumes outliers have larger variance. The latter case is more relevant for our setup, but we use blocks which are systematically placed (systematic outliers).
>
> Lastly, our paper also provides evidence for the claim that inliers can be modelled in a subspace: we introduce a model based on this assumption, and it performs better at outlier detection and data repair of systematic outliers (the error types in Table 1).

---

> ### Author Response · Authors · 2021-11-21
> **Answer to Review (Part 2)**
>
> We want to thank the reviewer for the time spent on our paper, thoughtful comments and questions, and the constructive feedback. We answer some of your questions and concerns below.
>
> **NOTE: additional references are in Answer Part-3 (i.e. for citations).**
>
> **Q2:
> The problem is clearly stated, but to me it is not evident what would be a practical application of this algorithm and a real use case. First of all, the fact that a labelled (trusted) data set is needed (albeit small) constrains the practical usage. Second, the use case of detecting images where some pixels are set to default values (e.g. from a deficient camera) would be dependent on a specific sensor. Finally, the application of watermarks is mentioned and makes more sense. However, the experimental evaluation does not include this more complex type of outliers.**
>
> **R2:**
>
> **1-** “...practical application of this algorithm and a real use case…”
>
> We decided to answer this question in a top-level comment (named **TPL-1**), for the benefit of all reviewers. So we kindly ask for the reviewer to see that comment.
>
> **2-** “...a labelled (trusted) data set is needed (albeit small) constrains the practical usage...”
>
> We understand the reviewer's concern about how much labelled data is needed. However, we disagree that this is an issue, given how small the trusted sets are. In our experiments we see that good results can be obtained by only labelling a few samples per systematic error: often as low as 5 or 10 samples. Further, as we mention in the introduction, we need to guide the model in understanding which instances are inliers / outliers. Unsupervised models cannot do this. Otherwise, systematic error repair for medium to large corruption levels is probably intractable in practice.
>
> Moreover, semi-supervised generative models (e.g. disentangled representation, or attribute manipulation) tend to use larger labelled sets -- see related work section. For instance, in our experiments, we showed that a recent SOTA baseline (CCVAE) needed more labelled data than us to solve this task. Looking at both CCVAE and related models [Ilse et al., 2020; Locatello et al., 2019] that use smaller labelled sets, we see that our model tends to need less labelled data.
>
> Secondly, there are currently several exploration tools for data visualization that allow for the user to explore and label the data much more easily. For instance, “FACETS” or “Know your Data” by Google, or the more popular “Tableau” could be used to easily label a few instances per systematic error. Further, given that data cleaning and data visualization is very common in setting up ML pipelines, even prior to model training, labelling a few samples per error should come as a byproduct of this.
>
> FACETS: https://ai.googleblog.com/2017/07/facets-open-source-visualization-tool.html
>
> Know your Data: https://knowyourdata.withgoogle.com/
>
> Tableau: https://www.tableau.com/
>
> **3-** “Use case of detecting images where some pixels are set to default values (e.g. from a deficient camera) would be dependent on a specific sensor.”
>
> Some types of systematic corruptions are common to different sensor types, e.g. if a network connection is disrupted, readings may be recorded as 0s. More importantly, if indeed the error is sensor specific, then only a few labels are needed per sensor error (e.g. 5 or 10 samples), which the user should be able to obtain during data visualization.
>
> **4-** “ application of watermarks is mentioned and makes more sense. However, the experimental evaluation does not include this more complex type of outliers.”
>
> We agree that more complex outliers like real world watermarks would make for a more compelling paper story. The main reason for not having included this is the lack of publicly available datasets for research, to our knowledge, which include ground-truth repairs and outlier labels (pixels and instances). Further, often watermarks are also injected synthetically [r2_ref_2][r2_ref_3]. However, as we mentioned in the top-level comment  **TPL-1**, we believe that the current synthetic errors are already difficult to repair, and similar patterns have been used in image inpainting.

---

> ### Author Response · Authors · 2021-11-21
> **Answer to Review (Part 3)**
>
> We want to thank the reviewer for the time spent on our paper, thoughtful comments and questions, and the constructive feedback. We answer some of your questions and concerns below.
>
> **Q3: There is an extensive appendix, which would be Ok. But the main paper misses some explanations and details that are only mentioned in the appendices (for example appendix E and F)**
>
> **R3:**
> We have made some improvements to the manuscript. These changes are highlighted in blue, and the summary can be found in a top-level comment to all reviewers. We’ll do our best to further clarify some of the explanations in the camera ready paper.
>
> ----------
> **REFERENCES used in this answer from Part-1 to Part-3** (not included in paper yet):
>
> **[r2_ref_1]:** “ Green Generative Modeling: Recycling Dirty Data using Recurrent Variational Autoencoders”, UAI 2017.
>
> **[r2_ref_2]:** Elharrouss, Omar, et al. "Image inpainting: A review." Neural Processing Letters 51, pages 2007–2028 (2020)
>
> **[r2_ref_3]:** Dekel, Tali, et al. "On the effectiveness of visible watermarks." Proceedings of the IEEE Conference on Computer Vision and Pattern Recognition. 2017.

---

### Official Review · Reviewer_Hvcm · 2021-11-09

**Correctness:** 4
**Technical Novelty And Significance:** 2
**Empirical Novelty And Significance:** 3
**Recommendation:** 6
**Confidence:** 3

**Main Review:**

Strength:
The problem is well motivated and clearly presented, very easy to follow. The formulation is very clear.
The performance of outlier removal is satisfactory on simple datasets.

Weakness:
The idea and formulation is a little incremental and similar to RVAE [Eduardo et.al.]. The main difference could be the semi-supervised mixture model.

Some thinking:
How correlated is the denoising quality and the downstream task, e.g., classification? could it be formulated together?
How is the model related to blind-source separation and ICA? Could the author gave some insight?

**Summary Of The Paper:**

The paper presents a Clean Subspace Variational Autoencoder (CLSVAE) model for systematic outlier detection. Building on VAE for latent variable modelling, the paper proposes a semi-supervised learning to infer the potential outlier.

The author made the following assumptions:
1. outliers exist therein, the inliers are still the majority;
2. Outliers are systematic with predictable recurring patterns;
3.  Compression hypothesis. Inlier data can be compressed further than outlier data.
4. Outliers are a combination of clean and dirty patterns.

On the model side, a mixture model is proposed parametrized by Bernoulli mixing weight and decoder p(x|z), while the encoder q(z|x) is applied for sampling latent variable z. A standard ELBO is formulated to maximise the log likelihood for both y known and unknown scenarios. For outlier repair, the author proposed to minimize the correlation between noise and clean signal.

Experimentally, the author evaluate two tasks: outlier detection, and automated repair on Frey-Faces3 , Fashion-MNIST (Xiao et al., 2017), Synthetic-Shapes.

**Summary Of The Review:**

The author presented CLSVAE to handle systematic outlier. The problem is well motivated and clearly presented, very easy to follow. The formulation is very clear. The performance of outlier removal is satisfactory on simple datasets.

The idea and formulation is a little incremental and similar to RVAE [Eduardo et.al.] with the main difference could be the semi-supervised mixture model.

As mentioned above, I wish the author could answer the two following questions:
1. How correlated is the denoising quality and the downstream task, e.g., classification? could it be formulated together?
2. How is the model related to blind-source separation and ICA? Could the author gave some insight?

---

> ### Author Response · Authors · 2021-11-21
> **Answer to Review (Part 1)**
>
> We want to thank the reviewer for the time spent on our paper, thoughtful comments and questions, and the constructive feedback. We answer some of your questions and concerns below.
>
>
> Rev1 (code: Hvcm).
>
> **Q1:
> The idea and formulation is a little incremental and similar to RVAE [Eduardo et.al.]. The main difference could be the semi-supervised mixture model.**
>
> **R1:**
> **There are similarities between CLSVAE and RVAE, however CLSVAE is not just a semi-supervised version of RVAE. The fundamental difference being that the noise process in RVAE is independent per pixel, whereas in CLSVAE the noise is correlated across pixels. This is what allows CLSVAE to model systematic errors, which is harder for RVAE.**
>
> **In more detail**, the RVAE model [Eduardo et.al. 2020] defines a two-component mixture model for each pixel (cell in tabular data, in RVAE paper), where the outlier component model has static parameters and is defined independently for each pixel. The inlier component model is defined by the decoder $p_\theta(\mathbf{x}|\mathbf{z})$ of a traditional VAE, and hence its latent space $\mathbf{z}$ only models the patterns associated with inlier instances (i.e. correlations between pixels associated with inliers). Ultimately, this means that the outlier component of RVAE is incapable of modelling errors that involve correlations between several anomalous pixels. This is the case for systematic errors where several pixels are affected simultaneously by a specific transformation (error type). Hence, the repair of systematic errors will be hindered, specially at medium to large corruption levels. However, random errors typically affect pixels independently, and so RVAE will work well.
>
> Conversely, CLSVAE defines a two-component mixture model for each datapoint (not pixel), where the decoder $p_\theta(\mathbf{x}|\mathbf{z})$ is used in both the inlier and outlier components. CLSVAE is able to model in its latent space (using $\mathbf{z}_d$) the correlation between pixels that the systematic error introduces, which is crucial for error repair. Being able to appropriately model the corruption pattern means that at repair time the model is able to “remove” its contribution from the reconstruction.
>
> **Q2:
> How correlated is the denoising quality and the downstream task, e.g., classification? could it be formulated together?**
>
> **R2:**
> **Generally, we would expect corruption (e.g. systematic errors) to have a negative impact on a downstream task**, e.g. classification or regression. We allude to this shortly in our introduction. Past literature confirms this exactly [Krishnan et al. (2016); Liu et al. (2020); Diakonikolas et al. (2018)], particularly systematic outliers. These works have shown that either repairing outliers (denoising) [Krishnan et al. (2016)], or removing outliers [Liu et al. (2020); Diakonikolas et al. (2018)], can improve performance of downstream tasks.
>
> *We do think that repair (denoising) and the downstream task can be formulated together.*
> An earlier example of this, for downstream linear models, is the work in [Krishnan et al. (2016)]. In our case, for CLSVAE, we could define a joint loss, or optimization procedure, using CLSVAE loss and the downstream task loss (assuming it is differentiable). In terms of architecture, there are two ways: (a) take the repaired instance $\tilde{x}$ from CLSVAE and “feed” it to the downstream task model; (b) after encoding the instance use latent representation $\mathbf{z}_c$ (inlier code), or perhaps [$\mathbf{z}_c$ ; $\mathbf{z}_d$], and feed it to the downstream task model. These are interesting directions for future work.

---

> > ### Author Response · Authors · 2021-11-21
> > **Answer to Review (Part 2)**
> >
> > **Q3:
> > How is the model related to blind-source separation and ICA? Could the author give some insight?**
> >
> > **R3:**
> > **We first note that ICA is a linear version of disentangled representation models (DMs)**, or alternatively disentanglement models. **We discuss DMs in Section 2 (last paragraph)**, as it pertains to our task. **The CCVAE baseline [Joy et al., 2020] is a non-linear example of a semi-supervised DM.** In the experiments (Section 5), we performed better than CCVAE because we did a better job at disentangling the inlier and error patterns.
> >
> > **Context:** From a blind-source separation perspective, repair (denoising) is the unmixing of the sources associated with inliers and those associated with the presence of systematic errors (outliers). ICA is frequently used to solve blind-source separation problems.
> >
> > - **Difference in Modelling (ICA vs CLSVAE):**
> >
> > (a) *ICA and DMs encapsulate pattern information about an attribute (source) $y$ in one variable or set of variables* ($z_y$). For the task of error repair (denoising), the y label defines inliers ($y=1$) and outliers ($y=0$). The remainder of $\mathbf{z}$ variables excluding $z_y$ (i.e. the code $z_{\setminus y}$ ) model unlabelled attributes (sources) of the data. Repairing is: maintain the $\mathbf{z}_{\setminus y}$ code and search for a value of $z_y$ that reconstructs an inlier (repair).
> >
> > (b) Conversely, *our proposal (CLSVAE) models inlier patterns ($y=1$) and error patterns ($y=0$) in separate variables -- not in the same variable(s) like ICA or DMs*. Specifically, for CLSVAE we have: ($y=1$) for inliers use $[\mathbf{z}_c ;  \mathbf{z}_\epsilon]$ where $\mathbf{z}_c$ captures inlier patterns, and $\mathbf{z}_\epsilon$ is uninformative noise around $\mathbf{0}$; ($y=0$) for outliers reuse the inlier code $\mathbf{z}_c$ and an error pattern code $\mathbf{z}_d$, where $[\mathbf{z}_c ; \mathbf{z}_d]$. Repairing is: just use $[\mathbf{z}_c ;  \mathbf{0}]$.
> >
> > - **Impact on Data Repair (Denoising):**
> >
> > (a) ICA and DMs:
> >
> > (1) Generally, ICA and DMs do not directly address the problem of data repair in their modelling, and thus are less efficient at this. Specifically, after model training, often we need to search the space $z_y$ for a value(s) that reconstruct a good repair. This requires extra effort, often with human intervention.
> >
> > (2) Even if a region of $z_y$ is known to model repair (y=1), e.g. CCVAE, often proper disentanglement (or independence) between inlier and error patterns is not assured. Leading to poor automatic repair.
> >
> > (b) Our model (CLSVAE):
> >
> > Unlike ICA or DMs, CLSVAE directly exploits the nature of the repair task by having separate representations for inlier patterns ($\mathbf{z}_c$) and error patterns ($\mathbf{z}_d$). In a corrupted dataset, most instances will be inliers and others will be outliers. The insight is the following: inliers will not need systematic error information ($\mathbf{z}_d$) for reconstruction, hence we use $\mathbf{0}$ (with noise) instead; outliers will need both information from $\mathbf{z}_c$ and $\mathbf{z}_d$ for reconstruction. This architecture naturally improves disentanglement between inlier and error patterns.
> > Also, this is why minimizing mutual information between $z_c$ (inliers) and $z_d$ (errors) leads to improved repair performance in difficult scenarios (see Section 4.4.).
> >
> > We will be happy to include parts of this discussion in the camera ready version (if accepted).

---

### Author Response · Authors · 2021-11-21
**Document Update**

We thank all reviewers for the time spent reviewing our submission, thoughtful questions and constructive feedback.

**All changes to the document are highlighted in blue.**

**Summary of changes:**

- **Added sentence** just before Section 4.1. to point out in Annex A there is a formulation for a standard VAE. This may help
some readers unfamiliar with VAEs.

- **Changed and clarified information in Section 4.2.**, about the variational model. Specifically, the $\pi_{\phi_y}(.)$ definition,
the purpose of *stop gradient* trick (stabilises optimization), and clarified a bit more why $q(y| \mathbf{z}_c, \mathbf{z}_d)$.

- **Annex E:** clarified some sentences.

- **Added Annex L** to *show empirically that corrupted data (with outliers) has larger variance than clean data (inliers only)*. We also cite two papers from literature that make a similar assumption on the variance of data corrupted by outliers (i.e. larger variance).

---

### Author Response · Authors · 2021-11-21
**Standard Experimental Practice, and Possible Applications (TPL-1)**

We thank all reviewers for their constructive feedback. This answer is placed here for the benefit of all.


We understand the reviewer's concerns, and we address this below.

+ **Synthetic Error Injection as Standard Experimental Practice**

As pointed out in Section 5 (Experiments), often prior research on data repair, or machine learning robust to corruption, has been carried out using synthetic error injection [Eduardo et al. (2020); Krishnan et al. (2016); Liu et al. (2020)]. For instance, in [Krishnan et al. (2016)] [REF_4] use similar corruptions (masking out pixels), but not for systematic error repair.
The main reason for this is precisely the lack of easily accessible public datasets for the data repair task, where for proper evaluation one needs the following targets: (a) labelling of all errors, either anomalous pixels or instances; (b) the ground-truth repairs, i.e. underlying inliers. This is especially frustrating given that data cleaning (outlier detection and repair) is quite common in machine learning pipelines (tabular or image data). Though these errors are synthetic, we believe them to be as hard to repair as several real examples.

Similarly, we can look at the related task of blind-inpainting in images [REF_1] [REF_2]. The goal of blind-inpaiting is to estimate which pixels are corrupted, or missing, and then infer the value of those pixels. However, unlike our setup of systematic errors, almost all of the models for blind-inpainting are trained using clean (uncorrupted) datasets [REF_1] [REF_2][REF_3], *a very important difference*. It is very common to have experiments mostly consisting of synthetic errors [REF_1] [REF_2], likely due to the lack of curated datasets for research. Specifically, we see that corruption related to: 1) image coding or transmission often involves errors as square blocks; 2) image restoration often involves inserting lines to emulate scratches, camera sensors failing, or watermarks; 3) other mask shapes can also be used to emulate object removal or occlusion, where often lines and geometric shapes are used.

+ **Possible Real-World Applications**

Medical imaging distortions (e.g. MRI, RX), restoration of images, or resolving issues related to bad image coding and transmission are possible applications. For instance, in medical imaging it is not uncommon to find corruptions that are: 1) similar to occlusion (by objects) as a result of dental or metal implants; 2) imaging sensor corruption (e.g. lines or objects) due to artifacts or sensor failure. We also believe that our model can be extended to tackle tabular data.

---------------------------
**References:**

**[REF_1]:** Elharrouss, Omar, et al. "Image inpainting: A review." Neural Processing Letters 51.2 (2020): 2007-2028.

**[REF_2]:** ​​Jam, Jireh, et al. "A comprehensive review of past and present image inpainting methods." Computer vision and image understanding 203 (2021)

**[REF_3]:** Dehaene, David, et al. "Iterative energy-based projection on a normal data manifold for anomaly localization." ICLR 2020

**[REF_4]:**  Green Generative Modeling: Recycling Dirty Data using Recurrent Variational Autoencoders, UAI 2017.

---

### Decision · Program_Chairs · 2022-01-20

**Decision:**

Reject

**Comment:**

This is a borderline paper with 2 marginally above and a marginally below acceptance recommendations. While the authors provided valid responses to some of the criticism, I still find some of the motivation and assumptions not sufficiently clear, theoretical and practical issues are mixed, and the validation on only synthetic data raises practical questions.